# V2X-Radar: A Multi-modal Dataset with 4D Radar for Cooperative Perception

**Lei Yang[1,2], Xinyu Zhang[1†], Jun Li[1], Chen Wang[3], Jiaqi Ma[4], Zhiying Song[1], Tong Zhao[1]**
**Ziying Song[5], Li Wang[1†], Mo Zhou[1], Yang Shen[1], Kai Wu[6],Chen Lv[2]**
[1]School of Vehicle and Mobility, Tsinghua University; [2]Nanyang Technological University
[3]CUMTB; [4]University of California, Los Angeles; [5]Beijing Jiaotong University; [6]ByteDance

## Abstract

Modern autonomous vehicle perception systems often struggle with occlusions and limited perception range. Previous studies have demonstrated the effectiveness of cooperative perception in extending the perception range and overcoming occlusions, thereby enhancing the safety of autonomous driving. In recent years, a series of cooperative perception datasets have emerged; however, these datasets primarily focus on cameras and LiDAR, neglecting 4D sensor used in single-vehicle autonomous driving to provide robust perception in adverse weather conditions. In this paper, to bridge the gap created by the absence of 4D Radar datasets in cooperative perception, we present V2X-Radar, the first large-scale, real-world multi-modal dataset featuring 4D Radar. V2X-Radar dataset is collected using a connected vehicle platform and an intelligent roadside unit equipped with 4D Radar, LiDAR, and multi-view cameras. The collected data encompasses sunny and rainy weather conditions, spanning daytime, dusk, and nighttime, as well as various typical challenging scenarios. The dataset consists of 20K LiDAR frames, 40K camera images, and 20K 4D Radar data, including 350K annotated boxes across five categories. To support various research domains, we have established V2X-Radar-C for cooperative perception, V2X-Radar-I for roadside perception, and V2X-Radar-V for single-vehicle perception. Furthermore, we provide comprehensive benchmarks across these three sub-datasets. We will release all datasets and benchmark codebase[1].

## 1 Introduction

Perception is critical in autonomous driving. While many single-vehicle perception methods [34; 33; 16; 27; 65; 51; 24; 23; 28; 15; 52; 31; 38; 32; 7; 30; 62] have emerged, substantial safety challenges persist due to occlusions and limited perception ranges. These issues occur because vehicles can only view their surroundings from a single perspective, resulting in an incomplete understanding of the scenario. This limitation hinders autonomous vehicles from achieving safe navigation and making optimal decisions. To address this challenge, recent studies have explored cooperative perception, where ego vehicles extend their perception range and overcome occlusions with assistance from other vehicles or roadside perception [49; 48; 50; 53; 37].

Recently, cooperative perception has attracted increasing attention, and several pioneering datasets have been released to bolster this research. For instance, datasets like OpenV2V [46], V2X-Sim [14], and V2XSet [45] are generated through simulations using CARLA [3] and SUMO [11]. In contrast, datasets such as DAIR-V2X [57], V2X-Seq [59], V2V4Real [44], and V2X-Real [39] are derived from real-world scenarios. However, a common limitation of these datasets is their exclusive focus

---

† Corresponding author.
[1]`https://github.com/yanglei18/V2X-Radar`

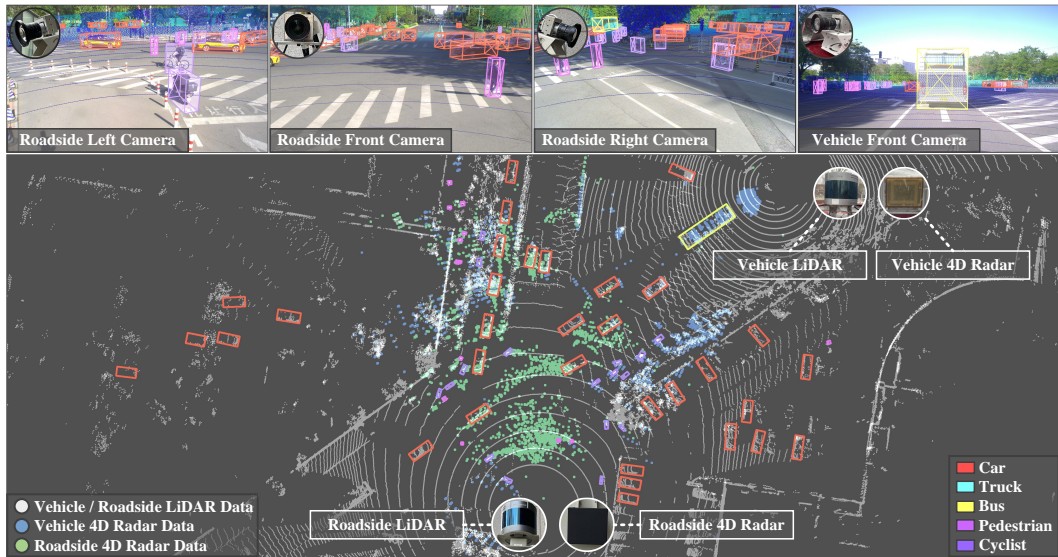

Figure 1: **A data frame sampled from the V2X-Radar dataset**. Each sample includes data from three sensors: (1) dense point clouds (gray points) from roadside and vehicle-side LiDAR; (2) sparse point clouds (green and blue points) with Doppler information from the roadside and vehicle-mounted 4D Radar; (3) RGB images (top row) from the vehicle-side camera and multi-view roadside cameras. All sensors are temporally and spatially synchronized. Each data frame is manually annotated with 3D boxes across five categories.

on camera and LiDAR sensors, neglecting the potential of 4D Radar. This sensor is considered advantageous for robust perception due to its exceptional adaptability to adverse weather conditions, as evidenced in single-vehicle autonomous driving datasets such as K-Radar [20] and Dual-Radar [63].

Table 1: **Comparisons of our proposed V2X-Radar with existing V2X datasets.** "C": Camera, "L": Lidar, "R": Radar, "V2V": vehicle-to-vehicle, "V2I": Vehicle-to-infrastructure.

| Dataset | Location | Sensor | V2X | Real | 4D Radar | Adverse Weather | Day&Night | #LiDAR | #Images | #4D Radar | #3D Box |
|---|---|---|---|---|---|---|---|---|---|---|---|
| OpenV2V [46] | Carla | L&C | V2V | ✗ | ✗ | ✗ | ✗ | 11K | 44K | 0 | 233K |
| V2X-Set [45] | Carla | L&C | V2V&I | ✗ | ✗ | ✗ | ✗ | 11K | 44K | 0 | 233K |
| DAIR-V2X [57] | China | L&C | V2I | ✓ | ✗ | ✗ | ✗ | 39K | 39K | 0 | 464K |
| V2X-Seq [59] | China | L&C | V2I | ✓ | ✗ | ✗ | ✗ | 39K | 39K | 0 | 464K |
| V2V4Real [44] | USA | L&C | V2V | ✓ | ✗ | ✗ | ✗ | 20K | 40K | 0 | 240K |
| TUMTraf-V2X[66] | DE | L&C | V2I | ✓ | ✗ | ✗ | ✓ | 2K | 5K | 0 | 29.38K |
| RCooper [8] | China | L&C | I2I | ✓ | ✗ | ✗ | ✗ | 30K | 50K | 0 | 310K |
| **V2X-Radar** | China | L&C&R | V2I | ✓ | ✓ | ✓ | ✓ | 20K | 40K | 20K | 350K |

To address the gap in the 4D Radar dataset for cooperative perception and to facilitate related studies, we present V2X-Radar, the first large-scale real-world multi-modal dataset featuring 4D Radar. This dataset includes a diverse range of scenarios, covering various weather conditions such as sunny, rainy, and snowy environments, and spans different times of the day, including daytime, dusk, and nighttime. The data was collected using a connected vehicle platform alongside an intelligent roadside unit, both equipped with 4D Radar, LiDAR, and multi-view cameras (Fig. 1). From over 15 hours of driving logs, we meticulously selected 50 representative scenarios for the final dataset. It comprises 20K LiDAR frames, 40K camera images, and 20K 4D Radar data, featuring 350K annotated bounding boxes across five object categories. To support a variety of research areas, V2X-Radar is further divided into three specialised sub-datasets: V2X-Radar-C for cooperative perception, V2X-Radar-I for roadside perception, and V2X-Radar-V for single-vehicle perception. Additionally, we provide comprehensive benchmarks of recent perception algorithms across these three sub-datasets. In comparison to existing real-world cooperative datasets, our V2X-Radar demonstrates two key strengths: (1) **More modalities**: The proposed V2X-Radar dataset includes three types of sensors: LiDAR, Camera, and 4D Radar, enabling further exploration into 4D Radar-related cooperative perception research. (2) **Diverse scenarios**: Our data collection covers diverse traffic densities, weather conditions (rain, fog, and snow), and times of day, focusing on complex intersections that pose challenges for single-vehicle autonomous driving. These scenarios include obstructed blind

spots that affect vehicle safety, providing rich and varied corner cases for advancing cooperative perception research.

Our contributions can be summarized as follows:

- We present V2X-Radar, the first large-scale real-world multi-modal dataset featuring 4D Radar for cooperative perception. The dataset is collected across diverse real-world scenarios with multiple sensor modalities, comprehensively covering various lighting and weather conditions to enable robust perception evaluation.

- We offer 20K LiDAR frames, 40K multi-view camera images, and 20K 4D Radar data, accompanied by 350K annotated bounding boxes across five object categories.

- Comprehensive benchmarks of recent perception algorithms are conducted across cooperative perception on V2X-Radar-C, roadside perception on V2X-Radar-I, and vehicle-side perception on V2X-Radar-V subsets.

## 2 Related Works

### 2.1 Autonomous Driving Datasets

Public datasets [1; 29; 46; 45; 14; 59; 57; 44; 39; 20; 63; 64; 35] have significantly accelerated the progress of autonomous driving in recent years. Single-vehicle datasets, such as KITTI [6], NuScenes [1], and Waymo [29], have notably furthered the development of single-vehicle perception. In terms of cooperative perception datasets, OPV2V [46] is the first dataset in this area, gathering data through co-simulation using CARLA [3] and OpenCDA [41; 42]. V2XSet [45] and V2X-Sim [14] explore V2X perception by employing synthesized data from the CARLA simulator [3]. Unlike simulated datasets, DAIR-V2X introduces the first real-world dataset for cooperative detection. V2X-Seq [59] extends sequences from DAIR-V2X [57] with track IDs, establishing a sequential perception and trajectory forecasting dataset. V2V4Real [44] presents the initial real-world V2V dataset gathered from two connected vehicles. V2X-Real [39] is a large-scale multi-modal multi-view dataset intended for V2X research, compiled using two connected automated vehicles and two intelligent roadside units. TUMTraf-V2X [66] is a multimodal, multi-view V2X cooperative perception dataset designed for 3D object detection and tracking in traffic scenarios. A common limitation among the cooperative perception datasets mentioned above is their narrow focus on camera and LiDAR sensors, overlooking the benefits of 4D Radar. 4D Radar is recognized for its superior adaptability to adverse weather conditions, as demonstrated by single-vehicle autonomous driving datasets such as K-Radar [20] and Dual-Radar [63]. Consequently, It's essential to create a multimodal dataset that integrates 4D Radar to enhance cooperative perception research.

### 2.2 Cooperative Perception

Cooperative perception aims to extend the range of perception and overcome occlusions in single-vehicle perception by leveraging shared information among connected agents. It can be classified into three main types based on fusion strategies: (1) Early Fusion, which involves transmitting raw sensor data for the ego vehicle to aggregate and predict objects; (2) Late Fusion, which integrates detection results for a consistent prediction; and (3) Intermediate Fusion, which shares and fuses intermediate high-level features. Recent state-of-the-art methods [36; 2; 43; 45; 9; 18; 36; 4; 58; 60; 5; 26] typically follow intermediate fusion, achieving the best trade-off between accuracy and bandwidth requirements. For instance, V2VNet [36] uses a graph neural network to refine features and perform joint perception and prediction iteratively. F-Cooper [2] introduces a pooling mechanism for identifying salient features. CoBEVT [43] introduced local-global sparse attention for improved cooperative BEV map segmentation. V2X-ViT [45] presented a unified vision transformer for multi-agent and multi-scale perception. HM-ViT [9] introduced a 3D heterogeneous graph transformer for camera and LiDAR fusion. FFNet [58] is a novel flow-based feature fusion framework that addresses temporal asynchrony in cooperative 3D object detection through feature flow prediction. HEAL [18] proposed an extensible collaborative perception framework to integrate new heterogeneous agents with minimal integration cost and performance decline.

# 3 V2X-Radar Dataset

To facilitate cooperative perception research using 4D Radar, we introduce V2X-Radar, the first extensive real-world multi-modal dataset featuring 4D Radar. In this section, we first describe the data acquisition in Sec. 3.1; then, we present the data annotation in Sec. 3.2; and finally, we delve into the diverse data distribution and dataset analysis in Sec. 3.3.

## 3.1 Data Acquisition

**Sensor Setup.** The dataset was collected using a connected vehicle-side platform (Fig. 2(a)) and an intelligent roadside unit (Fig. 2(b)). Both units are equipped with sensors, including 4D Radar, LiDAR, and multi-view cameras. Additionally, a GPS/IMU system is employed to achieve high-precision localisation, facilitating the initial point cloud registration between the vehicle-side and roadside platforms. A C-V2X unit is also integrated for wireless data transmission. The sensor layout configuration can be found in Fig. 2, while detailed specifications are listed in Tab. 2.

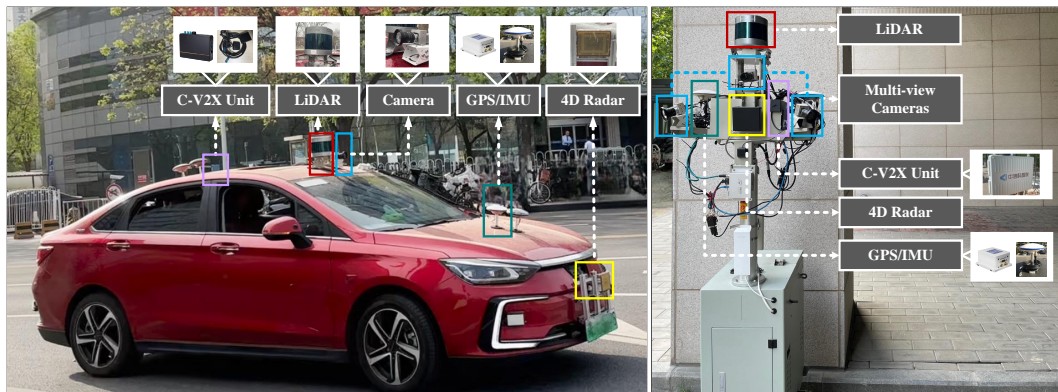

(a) Vehicle-side Platform    (b) Intelligent Roadside Unit

Figure 2: **The sensor configuration on the connected vehicle-side platform and the intelligent roadside unit.** a) the vehicle-side platform, and b) the intelligent roadside unit or infrastructure unit. Both are equipped with multi-modal sensors, including cameras, LiDAR, and 4D Radar, along with a C-V2X unit and a GPS/IMU system.

Table 2: **Sensor specifications in V2X-Radar dataset.** 'Infra.' means infrastructure or intelligent roadside unit.

| Agent | Sensor | Sensor Model | Details |
|---|---|---|---|
| Infra. | LiDAR | RoboSense RS-Ruby-80 ($\times$1) | 80 beams, $360°$ horizontal FOV, $-25°$ to $+25°$ vertical FOV |
| | Camera | Basler acA1920-40gc ($\times$3) | RGB, 1536$\times$864 resolution |
| | 4D Radar | OCULI EAGLE ($\times$1) | 79.0GHz, $-56°$ to $+56°$ horizontal FOV, $-22°$ to $+22°$ vertical FOV |
| | C-V2X Unit | VU4004 ($\times$1) | PC5/4G LTE/V2X Protocol |
| | GPS/IMU | XW-GI5651 ($\times$1) | 1000Hz update rate, Double-Precision |
| Vehicle | LiDAR | RoboSense RS-Ruby-80 ($\times$1) | 80 beams, $360°$ horizontal FOV, $-25°$ to $+25°$ vertical FOV |
| | Camera | Basler acA1920-40gc ($\times$1) | RGB, 1920$\times$1080 resolution |
| | 4D Radar | Arbe Phoenix ($\times$1) | 77GHz, $-50°$ to $+50°$ horizontal FOV, $-15°$ to $+15°$ vertical FOV |
| | C-V2X Unit | VU4004 ($\times$1) | PC5/4G LTE/V2X Protocol |
| | GPS/IMU | XW-GI5651 ($\times$1) | 1000Hz update rate, Double-Precision |

**Synchronization.** For cooperative perception datasets, synchronising sensors on both vehicle-side and roadside platforms using a uniform timestamp standard is crucial. To ensure consistency, all computer clocks are initially aligned with GPS time. Hardware-triggered synchronisation of LiDAR, cameras, and 4D Radar is then implemented using the Precision Time Protocol (PTP) and Pulse Per Second (PPS) signals. Subsequently, the closest LiDAR frames from the vehicle-side and roadside platforms are matched, and the camera and 4D Radar data are aligned with each corresponding LiDAR frame to create unified multi-modal data frames. Finally, the time difference between sensors across the two platforms is kept below 20 milliseconds for each sample.

**Sensor Calibration and Registration.** Through the sensor calibration process, we achieve spatial synchronisation of the camera, LiDAR, and 4D Radar. The intrinsic parameters of the camera are calibrated using a checkerboard pattern. In contrast, the LiDAR is calibrated relative to the camera by

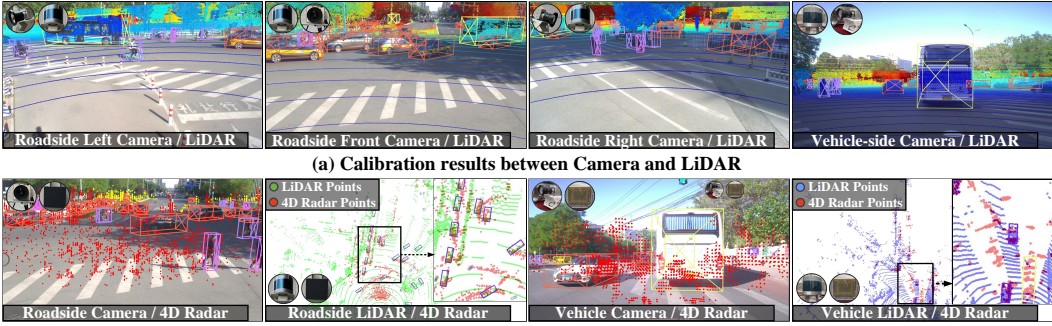

**(a) Calibration results between Camera and LiDAR**

**(b) Calibration results between 4D Radar and LiDAR / Camera**

Figure 3: **Visualization of calibration results.** a) The calibration results between the camera and LiDAR. b) The calibration results between the 4D Radar and LiDAR / Camera. The LiDAR points are projected onto the camera plane using the camera's intrinsic parameters and the camera-LiDAR extrinsics. Similarly, the 4D Radar points are transferred to the LiDAR coordinate system using the 4D Radar-LiDAR extrinsics. Additionally, these 4D Radar points are also mapped onto the camera plane by employing the camera's intrinsic parameters, along with the extrinsic parameters.

utilising 100 point pairs extracted from the point cloud and the corresponding camera image. The extrinsic parameters are derived by minimising the reprojection errors between the 2D-3D point correspondences. The calibration of the LiDAR with the 4D Radar is conducted by selecting 100 high-intensity point pairs located on corner reflectors. The results of this calibration are visually represented in Fig. 3. Vehicle-side LiDAR alignment with roadside LiDAR is achieved through point cloud registration, initially computed using RTK localisation and subsequently refined via CBM [25] and manual adjustments. The visualisation of the point cloud registration is shown in Fig. 4.

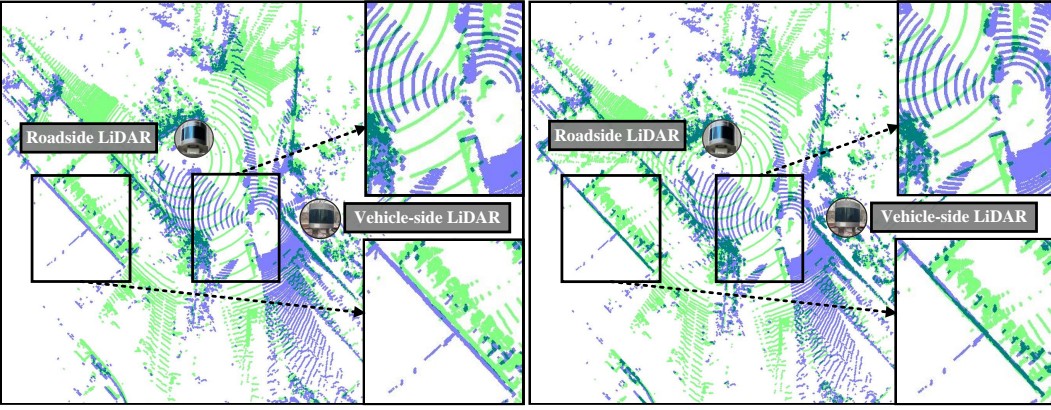

**(a) Initial point cloud registration based on RTK**    **(b) Refined point cloud registration with manual adjustment**

Figure 4: **Visualization of point cloud registration results.** a) Initial point cloud registration based on RTK localization. b) Refined point cloud registration with CBM [25] and manual adjustment. The blue points represent the point cloud from the vehicle-side LiDAR, while the green points indicate the point cloud from the roadside LiDAR.

**Data Collection.** Our data acquisition spanned over nine months, covering diverse environments such as university campuses, public roads, and closed testing parks, and ensuring comprehensive coverage across various weather conditions (sunny, rainy, foggy, and snowy) and times of day (daytime, dusk, and nighttime). In total, we collected 15 hours of driving data, comprising approximately 540K frames and encompassing numerous challenging intersection scenarios. From this dataset, we manually selected 40 representative sequences to construct V2X-Radar-C, each lasting between 10-25 seconds at a frequency of 10 Hz. Based on this, we further sampled 10 vehicle-only sequences to form V2X-Radar-V, and 10 infrastructure-only sequences to form V2X-Radar-I. Compared to the single-view configuration in V2X-Radar-C, both V2X-Radar-V and V2X-Radar-I feature a broader range of scenes. Altogether, the three subsets contain 20K LiDAR frames, 40K camera images, and 20K 4D radar samples. Further details are provided in the supplementary materials.

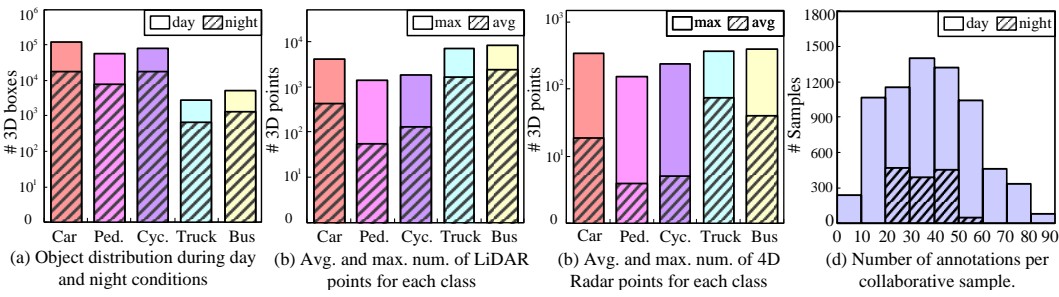

Figure 5: **Data analysis of our V2X-Radar dataset.** a) Distribution of objects during day and night conditions. b) Average and maximum number of LiDAR points within the 3D bounding box for each category. c) Average and maximum number of 4D Radar points within the 3D bounding box for each category. d) Number of annotations per collaborative sample. The vertical axes of sub-plots (a)-(c) use a log scale, whilst the vertical axis of sub-plot (d) employs a standard scale.

**Privacy Protection.** Before public release, the dataset undergoes a rigorous desensitization process. All privacy-sensitive elements, including road names, positioning data, road signs, license plates, and faces, are meticulously anonymized through a "model-based detection + manual verification" pipeline, in which a deep-learning model first blurs or masks sensitive regions, followed by frame-by-frame human inspection to ensure complete, compliant, and ethically responsible anonymization.

## 3.2 Data Annotation.

**Coordinate System.** Our dataset includes four types of coordinate systems: (1) LiDAR coordinate system, where the X, Y, and Z axes align with the front, left, and upward directions of the LiDAR. (2) Camera coordinate systems, in which the z-axis denotes depth. (3) 4D Radar coordinate system, with the X, Y, and Z axes oriented towards the right, front, and upward directions. (4) Global coordinate system, aligning with the LiDAR frame on the roadside platform.

**3D Bounding Boxes Annotation.** All data were annotated through an "auto-labeling + manual refinement" workflow: automated tools first generated initial labels, which were then carefully reviewed and corrected by human annotators, followed by multiple rounds of quality control to ensure accuracy and reliability. The annotation process encompasses vehicle-side, roadside, and collaborative annotations. Vehicle and roadside annotations were manually created within their respective LiDAR coordinate systems, while for collaborative annotation, the vehicle-side and roadside annotations were first aligned into a unified roadside LiDAR coordinate system and then matched using an IoU-based strategy to remove duplicates. Five object categories were annotated: pedestrian, cyclist, car, bus, and truck. Each object is represented as $(x, y, z, w, h, l, \theta)$, where $(l, w, h)$ represent the object's dimensions, $(x, y, z)$ indicate its location, and $\theta$ denotes its orientation.

## 3.3 Data Analysis

Fig. 5(a) illustrates the distribution of objects across five categories under both day and night conditions, with cars being the most prevalent in V2X-Radar, followed by cyclists and pedestrians, while trucks and buses are the least common. Fig. 5(b) displays the maximum and average number of LiDAR points within 3D bounding boxes for each category, suggesting that larger vehicles possess more 3D points than smaller pedestrians or cyclists. Fig. 5(c) reveals the density distribution of 4D Radar points within various objects' bounding boxes, reflecting a trend similar to that shown in Fig. 5(b). Lastly, Fig. 5(d) indicates that annotations per collaborative sample can reach up to 90, a significant increase compared to single-vehicle datasets such as KITTI [6] or nuScenes [1], emphasising how the integration of vehicle-side and roadside data enhances the comprehensiveness of environmental perception.

To better characterize the data composition, we performed a comprehensive statistical analysis on the V2X-Radar-V subset. The distribution results, shown in Fig. 6, demonstrate that the dataset covers a wide range of time-of-day conditions, including morning, afternoon, dusk, and night, as well as diverse weather scenarios such as sunny, fog, rain, and snow. Such balanced coverage captures both normal and adverse environmental conditions, providing a realistic and challenging benchmark for cooperative perception under varying illumination and visibility levels.

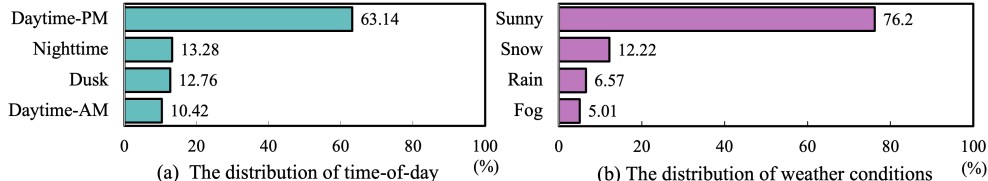

Figure 6: **Dataset distribution by time-of-day and weather conditions.** a) Distribution of data across different times of day (AM, PM, dusk, night). b) Distribution of data under various weather conditions (sunny, fog, rain, snow). The dataset spans multiple temporal and environmental scenarios, reflecting a balanced and realistic coverage of real-world driving conditions.

## 4 Tasks

V2X-Radar dataset comprises three sub-datasets: V2X-Radar-I, V2X-Radar-V, and V2X-Radar-C, which are designed for roadside 3D object detection, single-vehicle 3D object detection, and cooperative 3D object detection.

### 4.1 Single-agent 3D Object Detection

**Task Definition.** Single-agent 3D object detection involves two distinct tasks: roadside 3D object detection using the V2X-Radar-I sub-dataset, and vehicle-side 3D object detection employing the V2X-Radar-V sub-dataset. These tasks utilise sensors from either the intelligent roadside unit or the vehicle-side platform for 3D object detection, presenting the following challenges:

- **Single-modal Encoding:** The process of encoding a 2D image from the camera, a dense 3D point cloud from LiDAR, and a sparse point cloud with Doppler information from 4D Radar into a 3D spatial representation is essential for precise single-modal 3D object detection.
- **Multi-modal Fusion:** When fusing multi-modal information from various sensors, it is essential to account for (1) spatial misalignment, (2) temporal misalignment, and (3) sensor failure. Addressing these issues is crucial for achieving robust multi-modal 3D object detection.

**Evaluation Metrics.** The evaluation area covers [-100, 100] in x direction and [0, 100] in y direction from the ego-vehicle or roadside unit. Following the metrics in Rope3D [55], KITTI [6], we utilize the Average Precision (AP) at IoU thresholds of 0.5 and 0.7 as the evaluation metric.

**Benchmark Methods.** We conduct a comprehensive evaluation of various leading single-agent 3D object detectors based on different sensor inputs. Specifically, our evaluation encompasses LiDAR-centric techniques such as PointPillars [12], SECOND [47], CenterPoint [56], and PV-RCNN [22]; camera-dependent methods including SMOKE [17], BVDepth [13], BEVHeight [49], and BEVHeight++ [48]; and 4D Radar-based approaches like RPFA-Net [40] and RDIoU [21].

### 4.2 Cooperative 3D Object Detection

**Task Definition.** The cooperative 3D object detection task with the V2X-Radar-C sub-dataset aims to utilise sensors from both the vehicle-side platform and the intelligent roadside unit to execute 3D object detection for the ego-vehicle. In contrast to the previous single-agent perception, cooperative perception presents domain-specific challenges:

- **Spatial Asynchrony:** Localization errors can create discrepancies in the relative pose between the single-vehicle and the intelligent roadside unit, potentially leading to global misalignment when translating data from the roadside unit to the ego-vehicle coordinate systems.
- **Temporal Asynchrony:** Communication delays in the data transmission process can result in timestamp discrepancies between the sensor data from the single-vehicle platform and the intelligent roadside unit. This may cause local misalignment of dynamic objects when translating data within the same coordinate systems.

**Evaluation Metrics.** The evaluation area spans [-100, 100] metres in both the x and y directions relative to the ego vehicle. Similar to DAIR-V2X [57] and V2V4Real [44], we group various vehicle

types into the same class and concentrate solely on vehicle detection. The performance of object detection is evaluated using Average Precision (AP) at IoU thresholds of 0.5 and 0.7. Transmission costs are calculated through Average MegaBytes (AM). In line with previous studies [57; 44], we compare methods in two configurations: (1) Synchronous, ignoring communication delays. (2) Asynchronous, simulating the delay by retrieving roadside samples with the preceding timestamp.

**Benchmark Methods.** We provide comprehensive benchmarks for the following three fusion strategies in cooperative 3D object detection:

- **Late Fusion:** Each agent employs its sensors to detect 3D objects and shares the predictions. The receiving agent then applies NMS to generate the final outputs.

- **Early Fusion:** The ego vehicle collects all point clouds from itself and other agents into its own coordinate system, then proceeds with detection procedures.

- **Intermediate Fusion:** Each agent employs a neural feature extractor to obtain intermediate features; the encoded features are compressed and transmitted to the ego vehicle for cooperative feature fusion. We evaluate several prominent intermediate methods, including F-Cooper [2], V2XVIT [45], CoAlign [43], and HEAL [18], to establish a benchmark in this field.

## 5 Experiments

### 5.1 Implementation Details

For the dataset split, single-agent 3D object detection, which includes roadside and vehicle-side scenarios. The dataset is divided into train/val/test sets containing 7000, 1500, and 1500 frames, respectively. The selection of samples in this setup is entirely random. The cooperative 3D object detection dataset is divided into train/val/test sets with 30, 5, and 5 sequences. Notably, all experimental results are assessed using the validation set. For the detection areas, the single-agent detection techniques cover a range of [0, 100] m in the x-direction and [-100, 100] m in the y-direction relative to the ego vehicle or roadside unit. In contrast, cooperative 3D object detection encompasses a broader area, extending from [-100, 100] m in both the x and y directions relative to the ego vehicle. For the ground truth, LiDAR or camera methods utilize all the annotations. Considering the limited coverage area of 4D Radar, 4D Radar-based methods only use labels within its field of view for training and evaluation. For the feature extractor, all cooperative detection methods depend on PointPillar [12] to extract BEV features from LiDAR or 4D Radar point clouds and use LSS [10] to extract BEV features from images. All experiments are conducted using 4 RTX-3090 GPUs.

### 5.2 Benchmark Results

**Single-agent 3D Object Detection.** The benchmark results for the V2X-Radar-I and V2X-Radar-V sub-datasets under a homogeneous split are detailed in Tab. 3 and Tab. 4. The experimental results in these tables clearly demonstrate that LiDAR-based methods achieve the highest performance. Although 4D radar-based methods manage a relatively sparse point cloud, they still outperform camera-based methods. Camera-based methods, restricted by their inability to utilize depth information, are less effective than LiDAR and 4D Radar-based methods.

**Cooperative 3D Object Detection.** A quantitative comparison of representative methods on the V2X-Radar-C sub-dataset is shown in Tab. 5, and the main findings can be summarized as follows:

- In comparison to the benchmark of single-vehicle perception, all methods involving cooperative perception exhibit a significant performance enhancement, highlighting its essential role in improving single-vehicle perception.

- Compared to the results from the Sync setting, the introduction of communication delay in Async settings led to a considerable performance decline. As shown in Tab. 5 and Fig. 7, when subjected to transmission delay with a strict 0.7 IoU threshold, F-Cooper [2], V2X-ViT [45], CoAlign [19], and HEAL [18] using LiDAR point cloud experienced significant reductions. These findings underline the necessity of mitigating the effects of communication delay to ensure effective and robust cooperative perception.

Table 3: **Roadside 3D object detection benchmarks on V2X-Radar-I under homogeneous split.** The vehicle category includes car, bus and truck. "M" means modality, and "L", "C", and "R" denote LiDAR, Camera, and 4D Radar, respectively.

| Method | M | Vehicle (IoU = 0.7/0.5) ↑ | | | Pedestrian (IoU = 0.5/0.25) ↑ | | | Cyclist (IoU = 0.5/0.25) ↑ | | |
|---|---|---|---|---|---|---|---|---|---|---|
| | | Easy | Moderate | Hard | Easy | Moderate | Hard | Easy | Moderate | Hard |
| Pointpillars [12] | L | 78.00/88.69 | 71.24/81.16 | 71.24/81.16 | 53.87/69.61 | 52.94/67.09 | 52.94/67.09 | 80.01/87.48 | 72.67/78.60 | 72.67/78.60 |
| SECOND [47] | L | 81.61/91.06 | 74.34/83.47 | 74.34/83.47 | 57.56/74.64 | 55.39/72.27 | 55.39/72.27 | 80.53/87.68 | 74.06/80.62 | 74.06/80.62 |
| CenterPoint [56] | L | 86.44/94.04 | 78.98/84.26 | 78.98/84.26 | 67.90/84.74 | 65.39/81.35 | 65.39/81.35 | 90.26/92.91 | 82.87/85.51 | 82.87/85.51 |
| PV-RCNN [22] | L | 88.83/94.11 | 81.39/86.61 | 81.39/86.61 | 77.13/86.39 | 74.66/83.87 | 74.66/83.87 | 91.82/94.44 | 84.47/87.08 | 84.46/87.08 |
| SQDNet [61] | L | **89.12/95.10** | **81.48/86.94** | **81.48/86.94** | **78.02/88.47** | **74.79/84.19** | **74.79/84.19** | **92.13/95.43** | **84.53/87.59** | **84.53/87.59** |
| Fade3D [54] | L | 81.03/90.43 | 73.56/81.72 | 73.56/81.72 | 66.07/82.85 | 64.55/79.07 | 64.55/79.07 | 88.43/90.76 | 75.81/82.17 | 75.81/82.17 |
| SMOKE [17] | C | 22.05/58.43 | 20.72/56.36 | 20.69/56.31 | 8.26/25.64 | 7.68/24.36 | 7.63/24.30 | 12.50/38.29 | 11.28/36.40 | 11.24/36.36 |
| BEVDepth [13] | C | 45.01/69.25 | 42.23/66.81 | 42.21/66.75 | 30.64/61.46 | 29.13/59.11 | 29.10/59.05 | 39.85/68.71 | 38.52/67.05 | 38.43/67.02 |
| BEVHeight [49] | C | 47.91/72.45 | 45.53/69.48 | 45.49/67.44 | 32.08/64.06 | 29.78/59.79 | 29.68/59.74 | 42.97/71.63 | 41.34/69.11 | 41.30/69.07 |
| BEVHeight++[48] | C | 48.48/73.81 | 47.92/70.36 | 47.88/70.32 | 33.05/66.12 | 32.30/64.67 | 32.32/64.66 | 45.19/74.52 | 44.20/72.04 | 44.14/72.01 |
| RDIoU [21] | R | 61.38/80.03 | 54.89/72.72 | 54.89/72.72 | 43.82/72.03 | 42.11/69.65 | 42.11/69.65 | 40.74/67.31 | 36.68/60.83 | 36.68/60.83 |
| RPFA-Net [40] | R | 64.79/82.58 | 58.01/75.36 | 58.01/75.36 | 51.64/78.05 | 49.51/73.91 | 49.51/73.91 | 45.86/71.81 | 41.66/64.95 | 41.66/64.95 |

Table 4: **Single-vehicle 3D object detection benchmarks on V2X-Radar-V under homogeneous split.** The vehicle category includes car, bus and truck. "M" means modality, and "L", "C", and "R" denote LiDAR, Camera, and 4D Radar, respectively.

| Method | M | Vehicle (IoU = 0.7/0.5) ↑ | | | Pedestrian (IoU = 0.5/0.25) ↑ | | | Cyclist (IoU = 0.5/0.25) ↑ | | |
|---|---|---|---|---|---|---|---|---|---|---|
| | | Easy | Moderate | Hard | Easy | Moderate | Hard | Easy | Moderate | Hard |
| Pointpillars [12] | L | 75.66/83.52 | 68.80/77.07 | 68.80/77.07 | 41.89/46.34 | 38.16/43.18 | 38.16/43.18 | 78.63 /83.14 | 65.24/69.77 | 65.24/69.77 |
| SECOND [47] | L | 78.35/84.34 | 71.31/79.75 | 71.31/79.75 | 43.74/49.75 | 39.07/45.91 | 39.07/45.91 | 82.88/85.12 | 68.27/73.22 | 68.27/73.22 |
| CenterPoint [56] | L | 80.87/87.74 | 72.19/81.42 | 72.19/81.42 | 55.27/62.63 | 50.59/58.81 | 50.59/58.81 | 88.21/92.48 | 75.26/79.44 | 75.26/79.44 |
| PV-RCNN [22] | L | 88.27/89.12 | 79.38/84.31 | 79.38/84.31 | 67.04/69.81 | **58.83/65.78** | **58.83/65.78** | 89.48/93.49 | 78.01/81.07 | 78.01/81.07 |
| SQDNet [61] | L | **89.02/90.65** | **79.65/85.10** | **79.65/85.10** | **68.23/72.85** | 58.79/65.55 | 58.79/65.55 | **79.46/82.95** | **79.46/82.95** | **79.46/82.95** |
| Fade3D [54] | L | 79.24/85.77 | 70.64/78.95 | 70.64/78.95 | 57.82/65.92 | 51.88/60.94 | 51.88/60.94 | 84.95/88.29 | 69.93/75.80 | 69.93/75.80 |
| SMOKE [17] | C | 9.86/31.92 | 8.61/26.41 | 8.28/24.36 | 0.23/2.02 | 0.29 /1.89 | 0.29/1.89 | 0.39/7.23 | 0.37/4.96 | 0.37/4.13 |
| BEVDepth [13] | C | 16.91/41.63 | 15.47/39.68 | 15.02/37.83 | 9.92/29.98 | 8.51/27.76 | 8.49/27.72 | 12.18/47.20 | 9.46/39.34 | 9.30/39.15 |
| BEVHeight [49] | C | 16.58/40.32 | 15.32/39.15 | 14.08/37.30 | 9.49/28.50 | 8.48/26.46 | 8.39/26.57 | 9.58/44.56 | 7.35/35.04 | 7.26/34.91 |
| BEVHeight++[48] | C | 17.47/43.68 | 15.53/42.24 | 14.77/41.58 | 10.43/ 31.15 | 9.36/28.85 | 9.32/28.73 | 12.99/49.08 | 9.91/41.10 | 9.83/40.94 |
| RDIoU [21] | R | 41.11/72.67 | 29.03/54.27 | 28.37/52.02 | 10.72/28.59 | 9.97/26.88 | 9.84/26.78 | 14.74/44.57 | 10.81/31.15 | 10.67/30.91 |
| RPFA-Net [40] | R | 42.77/75.79 | 30.44/57.27 | 29.34/55.06 | 11.51/30.54 | 10.37/28.15 | 10.29/27.43 | 17.03/46.31 | 11.98/33.96 | 11.91/33.77 |

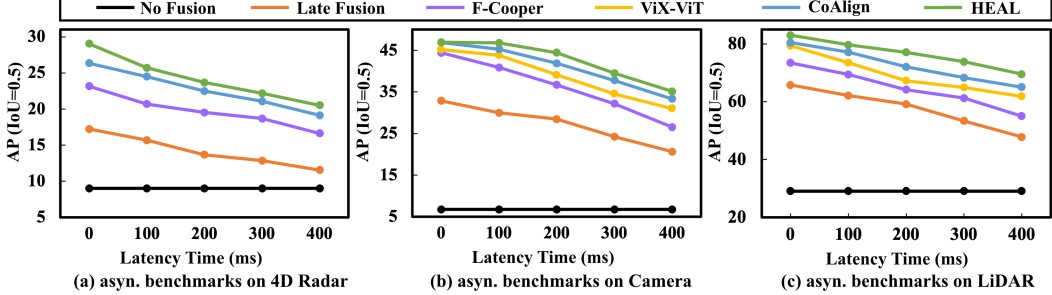

(a) asyn. benchmarks on 4D Radar   (b) asyn. benchmarks on Camera   (c) asyn. benchmarks on LiDAR

Figure 7: **Cooperative 3D object detection benchmarks on V2X-Radar-C under different transmission delay.** Each data point reflects the average three separate experiments.

**Adverse-weather Robustness through 4D Radar.** We conducted ablation studies on a subset of vehicle-side frames collected under adverse weather conditions (rain, fog, and snow), comparing LiDAR-only, 4D Radar-only, and LiDAR-4D Radar fusion models, as shown in Tab. 6. The results are as summarized as follows:

- While generally lagging behind LiDAR-only method under normal conditions, 4D Radar-only model outperforms LiDAR-only model by approximately 1–2% mAP in adverse weather conditions, highlighting its resilience to environmental degradation.

- The LiDAR-4D Radar fusion model consistently achieves the highest accuracy across both LiDAR-only and 4D Radar-only models, demonstrating the complementary sensing strengths of the two modalities, where LiDAR provides precise spatial geometry while 4D Radar contributes robustness under adverse conditions.

Table 5: **Cooperative 3D object detection benchmarks for vehicle category on V2X-Radar-C.** The vehicle category includes car, bus and truck. Sync. means synchronous setup ignoring communication delays. Async. implies asynchronous setup with a 100 ms delay.

| Method | M | Sync. (AP@IoU = 0.7 / 0.5) ↑ | | | | Async. (AP@IoU = 0.7 / 0.5) ↑ | | | |
|---|---|---|---|---|---|---|---|---|---|
| | | Overall | 0–30m | 30–50m | 50–100m | Overall | 0–30m | 30–50m | 50–100m |
| No Fusion | C | 1.00 / 6.76 | 2.04 / 9.41 | 0.18 / 5.16 | 0.01 / 2.27 | 1.00 / 6.76 | 2.04 / 9.41 | 0.18 / 5.16 | 0.01 / 2.27 |
| Late Fusion | C | 13.59 / 32.88 | 16.16 / 40.58 | 13.29 / 30.37 | 11.71 / 20.00 | 9.92 / 30.00 | 10.88 / 38.64 | 9.75 / 23.98 | 8.24 / 15.89 |
| F-Cooper | C | 15.56 / 44.43 | 23.22 / 61.97 | 10.98 / 31.24 | 4.15 / 15.38 | 14.25 / 40.90 | 23.27 / 57.38 | 6.86 / 29.07 | 3.62 / 13.74 |
| V2X-ViT | C | 15.22 / 45.19 | 20.13 / 57.74 | 11.20 / 35.31 | 9.30 / 24.84 | 13.50 / 43.85 | 17.24 / 56.28 | 11.84 / 33.02 | 8.37 / 20.08 |
| CoAlign | C | 24.26 / 46.89 | **36.49 / 63.70** | 12.75 / 32.77 | 11.36 / 23.17 | 21.36 / 45.29 | 28.86 / 57.47 | 12.36 / 33.71 | 11.71 / 22.17 |
| HEAL | C | **25.05 / 46.94** | 35.18 / 60.40 | **13.48 / 33.63** | **15.80 / 26.88** | **22.26 / 46.77** | **32.35 / 60.33** | **12.55 / 34.12** | **12.34 / 23.20** |
| No Fusion | L | 7.13 / 29.09 | 10.35 / 35.30 | 4.88 / 25.05 | 2.52 / 17.72 | 7.13 / 29.09 | 10.35 / 35.30 | 4.88 / 25.05 | 2.52 / 17.72 |
| Late Fusion | L | 39.37 / 65.75 | 50.92 / 80.59 | 31.59 / 58.64 | 18.54 / 31.62 | 34.51 / 62.10 | 46.85 / 70.32 | 23.31 / 56.86 | 15.12 / 23.41 |
| F-Cooper | L | 50.04 / 73.44 | 70.29 / 89.52 | 38.10 / 69.72 | 17.44 / 34.50 | 38.99 / 69.38 | 56.18 / 85.29 | 27.81 / 63.30 | 19.65 / 29.90 |
| V2X-ViT | L | 52.15 / 79.35 | 68.22 / 88.37 | 41.79 / 80.74 | 25.06 / 47.53 | 39.06 / 73.51 | 53.71 / 84.06 | 28.88 / 70.69 | 15.49 / 42.37 |
| CoAlign | L | 60.18 / 80.42 | 75.42 / 91.08 | 48.10 / 76.48 | 29.62 / 45.51 | 53.86 / 77.13 | 73.65 / 90.58 | 39.36 / 73.26 | 17.34 / 35.54 |
| HEAL | L | **67.57 / 83.00** | **82.76 / 92.19** | **57.70 / 80.51** | **34.79 / 51.93** | **57.76 / 79.66** | **74.62 / 90.19** | **42.33 / 74.18** | **21.10 / 47.15** |
| No Fusion | R | 2.69 / 9.02 | 4.59 / 13.39 | 0.93 / 5.97 | 0.22 / 1.28 | 2.69 / 9.02 | 4.59 / 13.39 | 0.93 / 5.97 | 0.22 / 1.28 |
| Late Fusion | R | 3.77 / 17.24 | 6.27 / 25.97 | 1.44 / 11.19 | 0.13 / 0.69 | 3.41 / 15.68 | 5.26 / 22.85 | 1.08 / 5.76 | 0.22 / 2.17 |
| F-Cooper | R | 6.84 / 23.16 | 11.80 / 34.98 | 2.74 / 16.87 | 0.38 / 2.05 | 6.37 / 20.70 | 12.37 / 30.86 | 2.42 / 12.84 | 0.16 / 1.74 |
| CoAlign | R | 11.46 / 26.34 | 18.01 / 38.46 | **5.75** / 16.55 | 0.28 / 2.83 | 11.12 / 25.49 | 16.35 / 38.69 | 3.25 / 14.14 | 0.22 / 2.34 |
| HEAL | R | **12.50 / 29.04** | **23.02 / 44.83** | 5.16 / **17.85** | **0.45 / 2.98** | **11.42 / 25.71** | **19.41 / 39.30** | **4.12 / 16.84** | **0.34 / 2.54** |

Table 6: **Ablation results under adverse weather conditions (rain, heavy fog, and snow).** 4D Radar-only method outperforms LiDAR-only model by 1-2% mAP in harsh conditions, and the fusion model consistently achieves the best results, demonstrating their complementary strengths.

| Method | Modality | Vehicle (IoU = 0.5) ↑ | | | Pedestrian (IoU = 0.25) ↑ | | | Cyclist (IoU = 0.25) ↑ | | |
|---|---|---|---|---|---|---|---|---|---|---|
| | | Easy | Moderate | Hard | Easy | Moderate | Hard | Easy | Moderate | Hard |
| Pointpillars [12] | LiDAR | 47.23 | 43.12 | 42.15 | 21.12 | 17.52 | 16.87 | 29.32 | 25.34 | 24.81 |
| | 4D Radar | 48.35 | 44.80 | 43.91 | 20.14 | 16.54 | 16.20 | 27.85 | 23.66 | 22.68 |
| M2-Fusion [56] | LiDAR + 4D Radar | **53.61** | **50.18** | **49.72** | **25.11** | **20.53** | **19.59** | **33.42** | **28.24** | **27.12** |

## 6 Conclusion

In this work, we introduced V2X-Radar, the first large-scale real-world multi-modal dataset featuring 4D Radar for cooperative perception. Our dataset focuses on challenging intersection scenarios and provides data collected under diverse times and weather conditions. Beyond releasing a dataset and benchmarks, our study revealed two key insights for the research community: (i) Severe performance degradation under asynchronous communication settings. This finding exposes a critical weakness in the delay robustness of current cooperative perception methods. (ii) The distinctive advantage of 4D Radar, which delivers reliable perception in adverse weather and serves as a valuable complement to LiDAR- and camera-based approaches. By releasing V2X-Radar, we not only fill the 4D Radar gap in cooperative perception research but also offer a platform for studying these challenges and validating solutions. We hope this dataset and benchmark will spark future work on delay-tolerant cooperative perception models, robust cross-modal fusion strategies.

**Broader Impacts.** Although the proposed benchmark covers various driving scenes, due to differences in sensor configurations, models trained on this dataset may not generalize well to other vehicle-side or roadside unit platforms, thus failing to ensure the safety of autonomous driving.

## 7 Limitation and Future Work

Our current dataset primarily focuses on 3D object detection, providing a foundation for studying cooperative perception under diverse sensing modalities and scenarios. However, it is limited in temporal coverage and task diversity, as it does not yet include sequential or predictive perception tasks. In future, we plan to extend the V2X-Radar dataset toward more comprehensive cooperative perception benchmarks. We will introduce multi-object tracking and trajectory prediction to capture temporal dynamics and enable spatiotemporal reasoning, and develop an Occupancy Prediction task formulated as voxel-level semantic segmentation with both dynamic and static classes. These extensions aim to enable richer scene understanding and advance V2X-Radar toward a complete world-model-based perception benchmark.

## Acknowledgment

This work was supported by the National Key Research and Development Program of China (2022YFB2503003), the National Natural Science Foundation of China (52221005, 62273198, 52072215, U1964203), the Beijing Natural Science Foundation (L241017, L243025). the Agency for Science, Technology and Research (A*STAR), Singapore, through the MTC Individual Research Grant (M22K2c0079), the CoE Dean's Interdisciplinary Grant at Nanyang Technological University, and the Ministry of Education, Singapore, through the Tier 2 Grant (MOE-T2EP50222-0002).

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
