# Supplementary for V2X-Radar: A Multi-modal Dataset with 4D Radar for Cooperative Perception

**Lei Yang[1,2], Xinyu Zhang[1]†, Jun Li[1], Chen Wang[3], Jiaqi Ma[4], Zhiying Song[1], Tong Zhao[1]**
**Ziying Song[5], Li Wang[1], Mo Zhou[1], Yang Shen[1], Kai Wu[6], Chen Lv[2]**
[1]School of Vehicle and Mobility, Tsinghua University; [2]Nanyang Technological University
[3]CUMTB; [4]University of California, Los Angeles; [5]Beijing Jiaotong University; [6]ByteDance

## A    Summary

This supplementary document is organized as follows:

- **Appendix B** provides detailed descriptions of the time synchronization mechanisms adopted in V2X-Radar, including synchronization within and across vehicle-side and roadside platforms.

- **Appendix C** presents the robustness evaluation of time synchronization under varying transmission delays, analyzing the impact of temporal asynchrony on cooperative perception performance.

- **Appendix D** presents the ablation study on the contribution of Doppler velocity information, showing that incorporating Doppler significantly enhances 4D Radar perception under adverse weather conditions.

- **Appendix E** compares the V2X-Radar single-agent subsets with representative autonomous driving datasets, highlighting their advantages in 4D Radar sensing, adverse-weather coverage, and multi-pass data collection.

- **Appendix F** introduces quantitative calibration metrics for LiDAR-Camera and LiDAR-4D Radar pairs, providing reprojection and alignment error analyses that demonstrate the dataset's high calibration precision.

- **Appendix G** showcases visual examples of diverse data collection scenarios, including corner cases for single-vehicle perception and variations across different times of day and weather conditions.

## B    Time Synchronization Details

Time synchronization enables the simultaneous sampling of data from the same scene by various sensors on both the vehicle platform and roadside units. For a collaborative perception dataset, achieving precise time alignment is critical not only among the diverse sensors within a single platform (be it the vehicle or the roadside unit) but also across sensors spanning both platforms.

The solution for time synchronization in our V2X-Radar is illustrated in Fig. 8. Typically, the vehicle platform and roadside unit align their sensors using a synchronization box that leverages the Precision Time Protocol (PTP). This synchronization box integrates Pulse Per Second (PPS) signals and Time of Day (ToD) data sourced from the GPS/IMU system. The process entails generating PTP timing signals to synchronize industrial computers, which subsequently facilitate hardware-triggered synchronization for LiDAR and 4D Radar. The PPS signals act as hardware triggers for camera data acquisition, utilizing rising edge pulses to ensure alignment between cameras, LiDAR, and 4D Radar within the same platform. To achieve time synchronization between the vehicle platform and the roadside unit, both systems receive identical GNSS signals from GPS satellites, ensuring precise clock alignment of the GPS/IMU systems across the two platforms.

---

† Corresponding author.
https://github.com/yanglei18/V2X-Radar

39th Conference on Neural Information Processing Systems (NeurIPS 2025) Track on Datasets and Benchmarks.

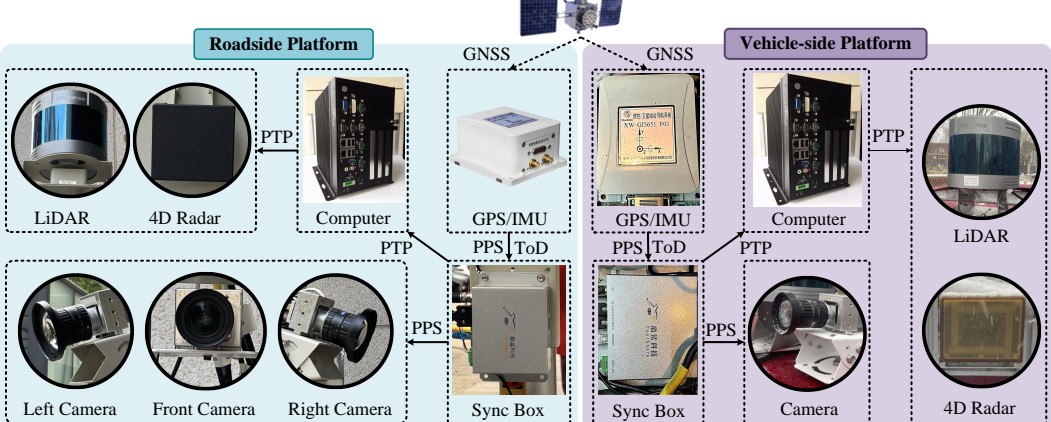

Figure 8: **The time synchronization solution for the connected vehicle-side platform and the intelligent roadside unit.** Both the vehicle platform and roadside unit are equipped with a computer, a time synchronization box, a GPS/IMU system, and various sensors, including LiDAR, 4D Radar, and cameras. Their GPS/IMU systems receive the same GNSS signals from GPS satellites.

Table 7: **Evaluation of time synchronization robustness on the V2X-Radar-C dataset under varying transmission delays.** Results are reported in AP@IoU=0.5. "M" denotes the sensing modality (L: LiDAR, C: Camera, R: 4D Radar). "AM" indicates the average transmission cost in MB.

| Method | M | Sync. (0ms) | Async.(100ms) | Async.(200ms) | Async.(300ms) | Async.(400ms) | AM (MB)↓ |
|---|---|---|---|---|---|---|---|
| No Fusion | L | 29.09 | 29.09 | 29.09 | 29.09 | 29.09 | **0.00** |
| Late Fusion | L | 65.75 | 62.10 | 59.15 | 53.33 | 47.76 | 0.003 |
| F-Cooper [2] | L | 73.44 | 69.38 | 64.15 | 61.26 | 55.01 | 1.25 |
| V2X-ViT | L | 79.35 | 73.51 | 67.23 | 64.92 | 61.87 | 1.25 |
| CoAlign [5] | L | 80.42 | 77.13 | 72.03 | 68.29 | 65.02 | 0.25 |
| HEAL [4] | L | **83.00** | **79.66** | **77.08** | **73.83** | **69.52** | 0.25 |
| No Fusion | C | 6.76 | 6.76 | 6.76 | 6.76 | 6.76 | **0.00** |
| Late Fusion | C | 32.88 | 30.00 | 28.45 | 24.23 | 20.64 | 0.003 |
| F-Cooper [2] | C | 44.43 | 40.90 | 36.72 | 32.19 | 26.55 | 1.25 |
| V2X-ViT | C | 45.19 | 43.85 | 39.14 | 34.55 | 31.03 | 1.25 |
| CoAlign [5] | C | 46.89 | 45.29 | 41.89 | 37.78 | 33.35 | 0.25 |
| HEAL [4] | C | **46.94** | **46.77** | **44.46** | **39.49** | **35.13** | 0.25 |
| No Fusion | R | 9.02 | 9.02 | 9.02 | 9.02 | 9.02 | **0.00** |
| Late Fusion | R | 17.24 | 15.68 | 13.68 | 12.85 | 11.56 | 0.003 |
| F-Cooper [2] | R | 23.16 | 20.70 | 19.52 | 18.69 | 16.63 | 1.25 |
| CoAlign [5] | R | 26.34 | 24.49 | 22.48 | 21.07 | 19.12 | 0.25 |
| HEAL [4] | R | **29.04** | **25.71** | **23.67** | **22.18** | **20.53** | 0.25 |

## C   Robustness Evaluation of Time Synchronization

To systematically assess the robustness of existing cooperative perception methods under temporal misalignment, we follow the time synchronization protocol adopted in DAIR-V2X [12] and V2V4Real [9], ensuring methodological consistency with established large-scale benchmarks. Specifically, we simulate varying transmission delays of 0 ms (synchronous), 100 ms, 200 ms, 300 ms, and 400 ms between vehicle-side and roadside sensors on the V2X-Radar-C dataset to comprehensively analyze how communication latency affects perception accuracy. These delay configurations are designed to emulate realistic transmission conditions in vehicular networks. As summarized in Tab. 7, the results show a clear and monotonic degradation trend as the delay increases. This consistent performance decline highlights the detrimental effect of temporal misalignment arising from transmission latency on multi-agent perception, indicating that even moderate desynchronization can substantially disrupt spatialâĂŞtemporal feature correspondence.

## D   Role of Doppler in 4D Radar Perception

To thoroughly assess the impact of Doppler velocity information, we conduct ablation studies comparing 4D Radar with and without Doppler measurements on a subset of vehicle-side frames captured under adverse weather conditions, including rain, fog, and snow. These challenging scenarios

are deliberately chosen to evaluate the robustness of radar-based perception when visual sensors are impaired by low visibility or specular noise. As summarized in Tab. 8, incorporating Doppler cues consistently improves detection performance across all object categories, delivering substantial gains for vehicles (+3.41 AP) and noticeable improvements for pedestrians and cyclists. The performance boost stems from Doppler's capacity to explicitly encode radial velocity, enabling better motion discrimination and mitigating background clutter interference. Compared with LiDAR, 4D Radar enhanced with Doppler not only narrows the performance gap but even surpasses LiDAR in severe weather (e.g., Vehicle: 43.12 vs. 44.80), underscoring its resilience to optical degradation. Overall, these results demonstrate that Doppler information is not merely an auxiliary signal but a critical sensing dimension that fundamentally enhances the reliability and environmental adaptability of radar perception.

Table 8: **Ablation results on Doppler's contribution to 4D Radar perception under adverse weather (rain, fog, and snow).** 4D Radar with Doppler yields consistent gains over the Doppler-free variant (+3.41 AP for vehicles) and even surpasses LiDAR under adverse weather (43.12 vs. 44.80).

| Modality | Veh. (IoU = 0.5)↑ | | | Ped. (IoU = 0.25)↑ | | | Cyc. (IoU = 0.25)↑ | | |
|---|---|---|---|---|---|---|---|---|---|
| | Easy | Moderate | Hard | Easy | Moderate | Hard | Easy | Moderate | Hard |
| LiDAR | 47.23 | 43.12 | 42.15 | **21.12** | **17.52** | **16.87** | **29.32** | **25.34** | **24.81** |
| 4D Radar w/o Doppler | 45.22 | 41.39 | 40.92 | 19.98 | 14.30 | 14.24 | 24.68 | 21.52 | 20.56 |
| 4D Radar w/ Doppler | **48.35** | **44.80** | **43.91** | 20.14 | 16.54 | 16.20 | 27.85 | 23.66 | 22.68 |

# E  Dataset Comparison from a Single-agent Perspective

Although V2X-Radar was originally built for cooperative perception, it also includes two single-agent subsets, V2X-Radar-I and V2X-Radar-V, that have unique strengths. They feature real 4D Radar sensing, broad coverage of challenging weather conditions, and repeated multi-pass recordings that enable accurate scene reconstruction and HD map generation. To highlight these strengths, we present a comparison table (Tab. 9) that focuses on single-agent datasets. Our subsets are compared with well-known autonomous driving datasets such as BDD100K [11], KITTI [3], nuScenes [1], Waymo [8], Rope3D [10], VOD [7], and K-Radar [6]. The results show that, while V2X-Radar was designed for cooperative use, its single-agent subsets provide higher-quality 4D Radar data, better performance in adverse weather, and richer multi-pass coverage, making them a strong resource for research on robust perception, scene reconstruction, and mapping.

Table 9: **Comparison of our single-agent dataset with existing autonomous driving datasets.** Although V2X-Radar is designed for cooperative perception, its single-agent subsets provide unique advantages in real 4D Radar sensing, adverse-weather coverage, and multi-pass data for scene reconstruction and HD mapping.

| Dataset | Location | Num. Data | LiDAR | Camera | 4D Radar | GPS / RTK | Day & Night | Adverse Weather | Multi-pass Coverage |
|---|---|---|---|---|---|---|---|---|---|
| BDD100K [11] | USA | 100K | ✗ | ✓ | ✗ | ✓ | ✓ | ✓ | ✗ |
| KITTI [3] | DE | 15K | ✓ | ✓ | ✗ | ✓ | ✗ | ✗ | ✗ |
| nuScenes [1] | Singapore | 40K | ✓ | ✓ | ✗ | ✓ | ✓ | ✗ | ✗ |
| Waymo [8] | USA | 230K | ✓ | ✓ | ✗ | ✗ | ✗ | ✗ | ✗ |
| Rope3D [10] | China | 40K | ✗ | ✓ | ✗ | ✗ | ✓ | ✓ | ✗ |
| VOD [7] | Netherlands | 8.7K | ✗ | ✓ | ✗ | ✗ | ✗ | ✗ | ✗ |
| K-Radar [6] | South Korea | 35K | ✓ | ✓ | ✓ | ✓ | ✓ | ✗ | ✗ |
| V2X-Radar-I | China | 20K | ✓ | ✓ | ✓ | ✓ | ✓ | ✓ | ✗ |
| V2X-Radar-V | China | 20K | ✓ | ✓ | ✓ | ✓ | ✓ | ✓ | ✓ |

# F  Quantitative Calibration Metrics

To quantitatively assess sensor calibration accuracy, we evaluate both LiDAR-Camera and LiDAR-4D Radar calibration errors.

For LiDAR-Camera calibration, the reprojection error (RPE) is used as the evaluation metric. A checkerboard target is employed to extract 3D corner points in the LiDAR frame and their corresponding pixel coordinates in the camera image. Using the estimated calibration parameters, the 3D points are projected onto the image plane, and the mean pixel-wise L2 distance between the projected and true corner locations is reported as the reprojection error (RPE).

For LiDAR-4D Radar calibration, the alignment error (AE) is used. Corner reflectors are employed to obtain paired 3D points in both LiDAR and Radar frames. Radar points are transformed into the LiDAR coordinate system based on the estimated extrinsics, and the mean Euclidean (L2) distance between the transformed Radar points and their corresponding LiDAR points is computed as the alignment error (AE).

As summarized in Tab. 10, the V2X-Radar dataset achieves LiDAR-Camera RPEs of 0.26-0.33 pixels (well below the acceptable 0.5 pixel threshold) and LiDAR-4D Radar AEs of 2.23-2.35 cm (within the acceptable < 5 cm range). These results demonstrate the dataset's high calibration precision and cross-sensor consistency across both vehicle-side and roadside platforms.

Table 10: **Calibration accuracy across different sensor pairs on both vehicle-side and roadside platforms.** LiDAR-Camera calibration is evaluated by the re-projection error (RPE, pixel), and LiDAR-4D Radar calibration by the alignment error (AE, cm). Lower values indicate more accurate extrinsic calibration.

| Platform | Sensor | RPE (pixel)↓ | AE (cm)↓ |
|---|---|---|---|
| Vehicle-side | LiDAR – CAM | 0.33 | / |
| | LiDAR – 4D Radar | / | 2.35 |
| Roadside | LiDAR – CAM_LEFT | **0.26** | / |
| | LiDAR – CAM_FRONT | 0.27 | / |
| | LiDAR – CAM_RIGHT | 0.30 | / |
| | LiDAR – 4D Radar | / | **2.23** |

# G  Data Collection Scenarios

Fig. 9 illustrates the diversity of data collection scenarios in the vehicle-side subset, spanning different times of day and a wide range of meteorological conditions. Specifically, Fig. 9(a) depicts temporal variations covering morning, afternoon, dusk, and nighttime scenes, while Fig. 9(b) presents weather diversity, including clear, foggy, rainy, and snowy environments. This comprehensive coverage enables the dataset to capture a broad spectrum of illumination and weather conditions, thereby supporting a more rigorous and comprehensive evaluation of perception robustness under diverse real-world challenges.

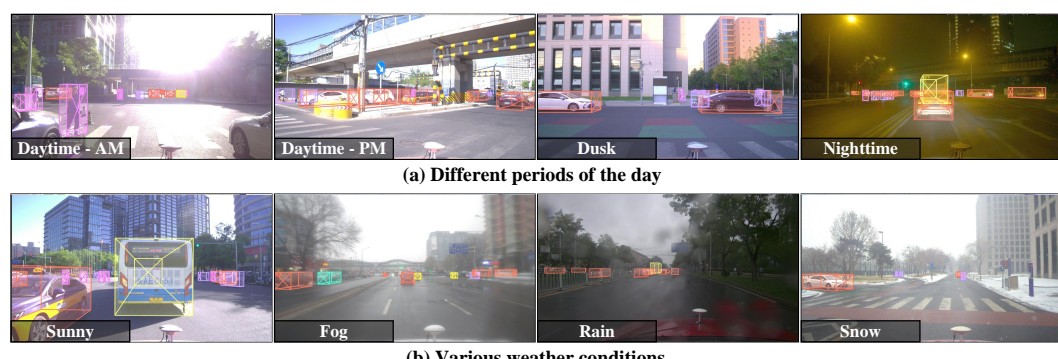

(a) Different periods of the day

(b) Various weather conditions

Figure 9: **Data collection across different periods and various weather conditions.** a) Different times of day, including daytime, dusk, and nighttime. b) Various weather conditions like sunshine, rain, fog, and snow.

Fig. 10 provides a comprehensive visualization of representative corner cases captured in the V2X-Radar dataset, illustrating the diverse and complex challenges encountered in real-world autonomous driving scenarios. These cases include occluded vehicles at public intersections, partially hidden cyclists at T-junctions, concealed pedestrians within restricted park areas, and obscured cyclists at campus crossings, all of which highlight the need for robust perception under diverse and safety-critical conditions.

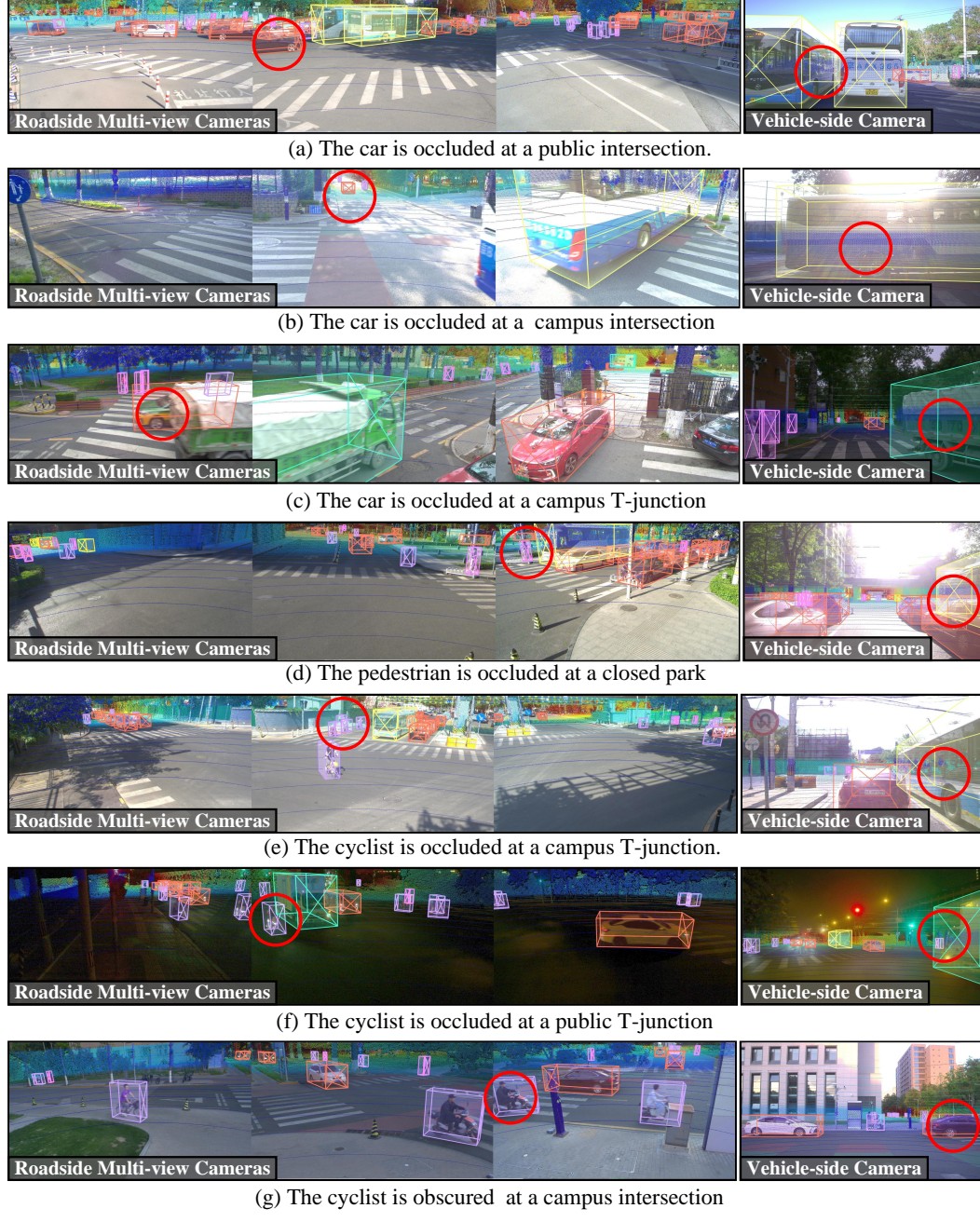

(a) The car is occluded at a public intersection.

(b) The car is occluded at a  campus intersection

(c) The car is occluded at a campus T-junction

(d) The pedestrian is occluded at a closed park

(e) The cyclist is occluded at a campus T-junction.

(f) The cyclist is occluded at a public T-junction

(g) The cyclist is obscured  at a campus intersection

Figure 10: **Diverse corner cases for single-vehicle autonomous driving.** Each case is depicted by roadside multi-view images on the left and the vehicle-side image on the right. We use a red circle to mark the occluded objects. The red circle highlights critical objects that are occluded from the vehicle-mounted perspective but remain observable from the roadside viewpoint.