# OpenReview forum: "V2X-Radar: A Multi-modal Dataset with 4D Radar for Cooperative Perception"
_NeurIPS.cc/2025/Datasets_and_Benchmarks_Track — NeurIPS 2025 Datasets and Benchmarks Track spotlight_

### Official Review · Reviewer_CafA · 2025-07-01

**Rating:** 5
**Confidence:** 4

**Summary:**

The authors introduce V2X-Radar, the first large-scale multimodal dataset featuring 4D Radar, LiDAR, and multi-view cameras, addressing the lack of 4D Radar in cooperative perception research. The dataset includes 20K LiDAR frames, 40K camera images, 20K 4D Radar data, and 350K annotated boxes, covering diverse weather, lighting, and challenging scenarios. It supports three sub-datasets—cooperative perception (V2X-Radar-C), roadside perception (V2X-Radar-I), and single-vehicle perception (V2X-Radar-V)—with comprehensive benchmarks to advance autonomous driving research.

**Dataset Code Accessibility:**

Yes

**Ethical Considerations:**

No, there are no or only very minor ethics concerns

**Final Justification:**

Overall, this paper demonstrates excellence in innovation, experimental design, and theoretical analysis, addressing most of the core issues raised in the review process.  I believe this paper meets the acceptance criteria. I recommend acceptance.

**Limitations Weaknesses:**

-The dataset appears to be collected in a specific geographic region, potentially limiting its generalizability to other regions with different road structures, traffic patterns, or environmental features.
-Consider adding more multimodal tasks (e.g., tracking) to further enhance the dataset’s utility.

**Strengths Contributions:**

-The overall structure of the paper is clear, and the language is fluent.
-The figures, tables, and captions are very informative, and I could clearly get the idea at first glance.
-V2X-Radar is the first large-scale dataset featuring 4D Radar for cooperative perception, highlighting the importance of 4D Radar in providing robust perception in adverse weather and complex environments. It addresses the lack of 4D Radar in existing datasets.

---

> ### Author Rebuttal · Authors · 2025-07-30
>
> We greatly appreciate your valuable comments and thoughtful questions. Below, we have provided detailed responses to address each of your concerns.
>
> ### **[W1]:** The dataset appears to be collected in a specific geographic region, which may limit generalizability to areas with different roads, traffic, or environments.
>
> Thanks for your insightful comment. We acknowledge that the dataset was primarily collected in a single metropolitan area. While geographically concentrated, the collection spans more than **10 distinct types of locations with different road structures** (including campuses, closed industrial parks, and public roads), capturing a broad spectrum of **traffic densities**, **diverse weather conditions**, and **both day and night periods**. This carefully curated diversity provides valuable variability that partially mitigates the limitation of regional focus and offers a strong foundation for research on cooperative perception in safety‑critical scenarios.
>
> ### **[W2]:** Consider adding more multimodal tasks (e.g., tracking) to further enhance the dataset’s utility.
>
> Thanks for your insightful comments. The V2X‑Radar dataset already **contains Tracking ID annotations**, providing a strong foundation for tasks such as tracking and prediction. In this rebuttal, we present an **initial tracking benchmark** (see table below) and will develop a more comprehensive, task‑agnostic baseline in future work to better support tracking, prediction, and broader V2X applications.
>
> | Method      | M  | AMOTA↑ | AMOTP↑ | sAMOTA↑ | MOTA↑ | MT↑   | ML↓   |
> |-------------|----|--------|--------|--------|-------|-------|-------|
> | No Fusion   | L  | 13.79  | 24.29  | 35.79  | 26.37 | 20.79 | 65.25 |
> | Late Fusion | L  | **29.78**  | **50.88**  | **64.78**  | **52.66** | **44.78** | **33.15** |
> | No Fusion   | C  | 3.60   | 9.40   | 21.70  | 10.63 | 8.33  | 89.33 |
> | Late Fusion | C  | 10.43  | 19.50  | 32.43  | 21.53 | 17.50 | 72.21 |
> | No Fusion   | R  | 4.22   | 12.50  | 22.22  | 13.85 | 9.22  | 85.02 |
> | Late Fusion | R  | 9.84   | 18.19  | 31.84  | 20.43 | 16.84 | 75.87 |

---

> > ### Comment · Reviewer_CafA · 2025-08-05
> >
> > Thanks for your careful response.

---

> > > ### Author Response · Authors · 2025-08-06
> > > **Thank You for Reading and Consideration**
> > >
> > > Dear Reviewer CafA,
> > >
> > > Thanks for your positive response! We truly appreciate your recognition and support of our work.
> > >
> > > Best regards,
> > >
> > > Authors

---

### Official Review · Reviewer_wjZP · 2025-07-02

**Rating:** 5
**Confidence:** 4

**Summary:**

This work proposed a dataset focusing on cooperative perception with multi-modal data: LiDAR, 4D-Radar and Camera. The dataset consists of 10K LiDAR, Radar +20K camera from Infra and 10K LiDAR, Radar, camera from Vehicles, which are collected across various time and weather conditions. The dataset is further divided into three subsets, and extensive baseline methods are evaluated on these subsets.

**Dataset Code Accessibility:**

Partly

**Dataset Code Comments:**

Sample data are accessible via the provided Kaggle link and the Google Drive link provided in the project page.

**Ethical Considerations:**

No, there are no or only very minor ethics concerns

**Final Justification:**

Thanks for the detailed rebuttal, which resolve most of my concerns. I raised my rating. Hope to see the revision in the next version.

**Limitations Weaknesses:**

- The baseline methods are not evaluated based on weather division. Since the main advantages of 4D mm Wave Radar is the robustness in adverse weather, the evaluations across weathers are important to demonstrate the advantages of 4D Radar in cooperative perception.
- Although the claimed perspective is for cooperative perception, two of the three subjects are for single agent task. It should also include the comparisons with datasets (e.g., K-Radar) collected for single agent task to better demonstrate the contribution.
- The full dataset is not accessible yet and not clear about the release data.

Minor weakness and questions
- There are many unreadable characters (e.g., Ã˚U) in both manuscript and supp.  Have a check.
- It is written that PTP is used for time synchronization. How about the trigger frequency for these sensors (the FPS of sensors are not described and may be different)?  10Hz as describe in the Data Collection part?
- It is not clear how many vehicle agents are used in the data acquisition. One? If so, the cooperative perception means cooperation between Infrastructure and Vehicle?
- Although the collection scenarios are well-categorized, the statistics of each scenario (number of frames day of time, weather) are not provided. It would be also better to illustrate the corresponding data of LiDAR and 4D Radar especially for adverse weather.
- Considering the FOV of camera and 4D Radar, does the vehicle agent move along the road in the FOV of Infra and towards the Infra? It would be nice to have the trajectory visualized.
- It is not clear how long does the data acquisition span in the time period to get data from different weathers. Or were they collected across different areas?
- For Table.5, it is better to conform with Table 3,4 in the order of IoU 0.5/0.7 or 0.7/0.5.

**Strengths Contributions:**

- The proposed dataset provided abundant data in various modalities and time/weather conditions, especially with 4D-mmWave Radar.
- The dataset is divided into three subsets for 3D object under different settings.
- The baseline methods are extensively evaluated.

---

> ### Author Rebuttal · Authors · 2025-07-30
>
> We greatly appreciate your valuable comments and thoughtful questions. Below, we have provided detailed responses to address each of your concerns.
>
> ### **[W1]:** Baselines lack weather‑specific evaluation, obscuring 4D Radar’s advantage in adverse conditions.
>
> Thanks for your valuable suggestions. In response, we conducted additional **ablation studies** on data collected under **adverse weather conditions** (rain, heavy fog, and snow), comparing **LiDAR-only**, **4D Radar-only**, and **LiDAR–4D Radar fusion** models, as shown in the table below. Our findings are as follows:
>
> 1. 4D Radar, while largely behind LiDAR in normal conditions, outperforms LiDAR by ~1–2% mAP under adverse weather, highlighting its robustness to environmental degradation.
>
> 2. The fusion model consistently delivers the best performance across all scenarios, demonstrating the complementary strengths of LiDAR and 4D Radar.
>
> These results quantify 4D Radar’s contribution and provide clear evidence of its critical role in improving perception robustness in adverse weather.
>
> | Method      | Modality         | Veh.(IoU = 0.5)↑ |          |      | Ped.(IoU = 0.25)↑ |          |      | Cyc.(IoU = 0.25)↑ |          |      |
> | ----------- | ---------------- | ------------------- | -------- | ---- | ----------------------- | -------- | ---- | -------------------- | -------- | ---- |
> |             |                  | Easy                | Mod. | Hard | Easy                    | Mod. | Hard | Easy                 | Mod. | Hard |
> | Pointpillar | LiDAR            | 47.23                  | 43.12                      | 42.15                  | 21.12                      | 17.52                         | 16.87                     | 29.32                  | 25.34                       | 24.81                  |
> | Pointpillar | 4D Radar         | 48.35                  | 44.80                      | 43.91                  | 20.14                      | 16.54                         | 16.20                     | 27.85                  | 23.66                       | 22.68                  |
> | M2-Fusion   | LiDAR + 4D Radar | **53.61**                  | **50.18**                      | **49.72**                  | **25.11**                      | **20.53**                         | **19.59**                     | **33.42**                  | **28.24**                       | **27.12**                  |
>
> ### **[W2]:** Although positioned as cooperative perception, comparisons with single‑agent datasets are needed to strength its contribution.
>
> Thanks for your comments. We have added a new comparison table focusing on single‑agent datasets, comparing our **V2X‑Radar‑I** and **V2X‑Radar‑V** subsets with representative benchmarks such as BDD100K [A], KITTI [B], nuScenes [C], Waymo [D], Rope3D [E], VOD [F], and K‑Radar [G]. While V2X‑Radar is designed for cooperative perception, our single‑agent subsets outperform existing datasets in real **4D Radar use**, **adverse‑weather**, and **multi‑pass collection**—repeatedly scanning the same areas to support scene reconstruction and HD map generation.
>
> | Dataset       | Num. Data | Location     | LiDAR | Camera | 4D Radar | GPS / RTK | Day&Night | Adverse Weather | multi-pass coverage |
> |--------------|-----------|-------------|-------|--------|----------|-----------|-----------|----------------|---------------------|
> | BDD100K      | 100K      | USA         | ❌    | ✅     | ❌       | ✅        | ✅        | ✅             | ❌                  |
> | KITTI        | 15K       | DE          | ✅    | ✅     | ❌       | ✅        | ❌        | ❌             | ❌                  |
> | nuScenes     | 40K       | Singapore   | ✅    | ✅     | ❌       | ✅        | ✅        | ❌             | ❌                  |
> | Waymo        | 230K      | USA         | ✅    | ✅     | ❌       | ❌        | ✅        | ❌             | ❌                  |
> | Rope3D       | 40K       | China       | ❌    | ✅     | ❌       | ❌        | ✅        | ✅             | ❌                  |
> | VOD          | 8.7K      | Netherlands | ❌    | ✅     | ❌       | ✅        | ❌        | ❌             | ❌                  |
> | K-Radar      | 35K       | South Korea | ✅    | ✅     | ✅       | ✅        | ✅        | ✅             | ❌                  |
> | V2X-Radar-I  | 20K       | China       | ✅    | ✅     | ✅       | ✅        | ✅        | ✅             | ❌                  |
> | V2X-Radar-V  | 20K       | China       | ✅    | ✅     | ✅       | ✅        | ✅        | ✅             | ✅                  |
>
> ### **[W3]:** The full dataset is not accessible yet and not clear about the release data.
>
> Thank you for your insightful question. We have released the V2X‑Radar **mini‑split** to help researchers explore the dataset structure and reproduce benchmarks, and the **full V2X‑Radar‑V**  also has been open‑sourced for direct use. The **remaining data** is undergoing **final privacy filtering** to meet legal and ethical standards and will be **released immediately after completion**.
>
> ### **[W&Q1]:** There are many unreadable characters (e.g., Ã˚U) in both manuscript and supplementary material .
>
> Thanks for your careful observation. The unreadable characters (e.g., “Ã˚U”) in **Figure 1**, **Table 2**, and **Section B** in supplementary material  were caused by LaTeX encoding issues. We have corrected all affected content and standardized the encoding to prevent recurrence in the final version.
>
> ### **[W&Q2]:** PTP is used for time sync, but are the sensors all triggered at 10 Hz, or do their FPS differ?
>
> Thank you for raising this point. All sensors (LiDAR, Cameras, 4D Radar) are hardware‑triggered via a shared GNSS/IMU clock at **10 Hz**, ensuring precise temporal alignment as noted in the Data Collection section.
>
> ### **[W&Q3]:** How many vehicles were used—one? If so, is “cooperative perception” only vehicle–infrastructure?
>
> Thanks for your question. Data were collected with one vehicle agent and a roadside unit, so the cooperative perception here refers specifically to vehicle–infrastructure (V2I) cooperation.
>
> ### **[W&Q4]:**  Scenarios are well categorized, but frame counts by time / weather are missing, and LiDAR/4D Radar data in adverse weather should be shown.
>
> Thanks for your valuable suggestions.
>
> 1. We have compiled detailed statistics for different scenarios, including **time of day** (day/night) and **weather conditions**, and summarized them in the following table.
>
> 2. Due to rebuttal media limits, we cannot show LiDAR and 4D Radar data under adverse weather now, but will include these visualizations in the final supplementary materials.
>
> | Time of Day      | Daytime-AM | Daytime-PM | Dusk   | Night|
> |------------------|------------|------------|--------|------------|
> | Proportion(%)  | 10.42      | 63.14      | 12.76  | 13.28      |
>
> | Weather Conditions | Sunny | Fog  | Rain | Snow  |
> |--------------------|-------|-----|------|-------|
> | Proportion(%)   | 73.2  | 5.01| 6.57 | 15.22 |
>
> ### **[W&Q5]:** Considering the camera and 4D Radar FOV, does the vehicle move within Infra’s view and approach it? A trajectory visualization would help.
>
> Thanks for your suggestion. For the selected annotated data, the vehicle was driven **within the infrastructure sensors’ FOV and toward the RSU** to ensure overlapping coverage. While trajectory plots cannot be included here due to rebuttal media limits, **they will be provided in the final supplementary materials**.
>
> ### **[W&Q6]:** It’s unclear how long data collection spanned to cover different weathers—or if it came from multiple areas.
>
> Thanks for your request for clarification. Our data acquisition spanned over **nine months**, ensuring coverage across a wide range of weather conditions (sunny, rain, fog, and snow). The collection sites included **campuses**, **public roads**, and **closed parks**, providing diverse areas.
>
> ### **[W&Q7]:** For Table.5, it is better to conform with Table 3,4 in the order of IoU 0.5/0.7 or 0.7/0.5.
>
> Thanks for your valuable suggestion. We will revise **Table 5** to align with **Tables 3 and 4** by presenting the results in a consistent IoU order (**IOU 0.7/0.5**) in the final version.
>
> ### **Reference:**
>
> [A] Yu F, Chen H, Wang X, et al. Bdd100k: A diverse driving dataset for heterogeneous multitask learning[C]//Proceedings of the IEEE/CVF conference on computer vision and pattern recognition. 2020: 2636-2645.
>
> [B] Geiger A, Lenz P, Urtasun R. Are we ready for autonomous driving? the kitti vision benchmark suite[C]//2012 IEEE conference on computer vision and pattern recognition. IEEE, 2012: 3354-3361.
>
> [C] Caesar H, Bankiti V, Lang A H, et al. nuscenes: A multimodal dataset for autonomous driving[C]//Proceedings of the IEEE/CVF conference on computer vision and pattern recognition. 2020: 11621-11631.
>
> [D] Sun P, Kretzschmar H, Dotiwalla X, et al. Scalability in perception for autonomous driving: Waymo open dataset[C]//Proceedings of the IEEE/CVF conference on computer vision and pattern recognition. 2020: 2446-2454.
>
> [E] Ye X, Shu M, Li H, et al. Rope3d: The roadside perception dataset for autonomous driving and monocular 3d object detection task[C]//Proceedings of the IEEE/CVF Conference on Computer Vision and Pattern Recognition. 2022: 21341-21350.
>
> [F] Palffy A, Pool E, Baratam S, et al. Multi-class road user detection with 3+ 1d radar in the view-of-delft dataset[J]. IEEE Robotics and Automation Letters, 2022, 7(2): 4961-4968.
>
> [G] Paek D H, Kong S H, Wijaya K T. K-radar: 4d radar object detection for autonomous driving in various weather conditions[J]. Advances in Neural Information Processing Systems, 2022, 35: 3819-3829.

---

> > ### Comment · Reviewer_wjZP · 2025-08-04
> >
> > Thanks for the detailed rebuttal, which resolve most of my concerns. I raised my rating. Hope to see the revision in the next version.

---

> > ### Author Response · Authors · 2025-08-05
> > **Thank You for Reading and Consideration**
> >
> > Dear Reviewer wjZP,
> >
> > Thank you for your positive response and for raising your score! We are pleased that our clarifications addressed your concerns and will ensure the corresponding clarifications are reflected in the final version.
> >
> > Best regards,
> > Authors

---

### Official Review · Reviewer_o696 · 2025-07-02

**Rating:** 5
**Confidence:** 4

**Summary:**

The submission presents a new cooperative object detection dataset for driving tasks. It was recorded in China and provides various data modalities, comprising 4D radar, LiDAR, camera, GPS, and IMU data under diverse weather conditions. Alongside the data recorded by the vehicle carrying the sonsor set further data is recorded by an infrastructure element, equipped with a sensor set. 3D bounding boxes are manually annotated together with a class label. Different state-of-the-art detection methods are compared on the new dataset. The authors identify a clear problem for future work: asynchronicity leads to significant performance losses in cooperative perception.

**Additional Feedback:**

Questions:
- Current annotations contain only 5 classes. Are you going to expand the annotation pool or even further, creating more fine-grained annotations such as semantic/instance segmentation?
- Fig. 5: why are you using a log scale for (a)-(c) but a linear scale for (d)? This lets the ratio of night samples vs. day samples seem much larger than it really is.
- Fig. 5: are there 10^5 overall car detections collected of which ~10^4 were collected at night time? Or are there 10^5 daytime detections and ~10^4 nighttime detections?
In case that option one is correct: wouldn't be more helpful if the bars didn't overlap but were plotted next to each other?
- Were all annotations labeled manually? Was there any technical support by the labeling interface? Were autolabels used and refined by the annotators?
- How are the data anonymized? Is this done manually? Is a learned model or heuristic used? If so - which one?

Suggestions:
- The methods selected for benchmarking LiDAR object detection are relatively older.  Evaluation on more recent works will strengthen the experimental section.
- A visualization of the coordinate systems and their relation to each other could help the reader to understand the setup.
- To make the conclusion less redundant: this is not abstract or intro 2.0. Don't repeat too many specific numbers and rather present additional insights that were gained when gathering the data and running the benchmark experiments. For example it is very interesting how severely the performance degrades when using the asynchronous setting. You identified this issue and you present the problem together with a dataset to work on it to the research community. Repeat this important insight in the conclusion.

**Dataset Code Accessibility:**

Partly

**Dataset Code Comments:**

Code is already available as well as a mini split of the data. The full data release is still pending.

**Ethical Considerations:**

No, there are no or only very minor ethics concerns

**Final Justification:**

Authors have resolved my concerns in the rebuttal phase. Thus, I will raise the rating and suggest the acceptance for the paper.

**Limitations Weaknesses:**

- Although some visualizations of calibration results are provided, the paper does not include quantitative metrics (e.g., radar–LiDAR alignment error, reprojection error). These are important for evaluating dataset reliability and for users seeking to understand uncertainty in perception.
- While the authors argue that 4D Radar is important, especially in adverse weather conditions, the benchmarks do not directly quantify its benefit (e.g., comparison of multi-sensor-fusion models with/without 4D Radar in adverse weather scenarios). A clear ablation would better support the claim.
- In Section 5.1, it is stated that “The selection of samples in this setup is entirely random.” It is unclear whether random selection is performed on entire sequences or individual frames. If frames are sampled independently, then domain gaps between splits may be small. This limits the benchmark’s ability to test generalization.
- The labeling and anonymization processes are not described in detail. (See questions below.)
- The plots in Fig. 5 are confusing to read. (See questions below.)
- Tables could benefit from highlighting the data
- The conclusion is very redundant with the abstract and intro.
- Minor mistakes:
	-  Fig. 1 caption: "the vehicleâ˘A ´ Zs camera"
	- Table 2: RGB, 1536ÃU˚ 864 resolution; RGB, 1920ÃU˚ 1080 resolution
	- Fig. 5 caption: "standard scale" should be "linear scale"
l. 186, 206, 230: Units missing. Use consistent unit annotations and either write "meters" or rather use "m" everywhere. Spelling "meters" not "metres".

**Strengths Contributions:**

- Good novelty: the dataset addresses a clear gap in current public datasets by including 4D Radar, which is a critical sensor for robust perception under adverse conditions.
- Flexible setup: different splits facilitate working on different perception tasks; namely single-vehicle roadside, and cooperative object detection.
- New benchmark and extensive evaluation of state-of-the-art methods on that: the paper provides strong baselines for both single-agent and cooperative detection setups across multiple modalities and multi-agent fusion strategies.
- The authors clearly explain the multi-sensor synchronization and calibration process, which is critical for usability and reproducibility.
- Identify a problem of current approaches and provide it to the community together with the data to work on a solution
- Code available and mini split of the data released
- Well-written and organized with very few spelling mistakes (see below) and well understandable

---

> ### Author Rebuttal · Authors · 2025-07-30
>
> We greatly appreciate your valuable comments and thoughtful questions. Below, we have provided detailed responses to address each of your concerns.
>
> ### **[W1]:** The paper shows calibration visualizations but lacks quantitative metrics.
>
> Thanks for your comments on calibration evaluation. We use two key metrics to assess calibration results:
>
> 1. **LiDAR–Camera** calibration error is measured by **re‑projection error (RPE)**. Using a checkerboard, we obtain 3D corner points in the LiDAR frame and their pixel coordinates in the camera image. With the estimated calibration parameters, the 3D points are projected onto the image plane, and the mean pixel‑wise L2 distance between projected and true corners is reported as the RPE.
>
> 2. **LiDAR–4D Radar** calibration error is measured by the **alignment error (AE)**. We use corner reflectors to capture 3D points in both LiDAR and Radar frames. Radar points are transformed into the LiDAR frame using the estimated calibration parameters, and the mean Euclidean (L2) distance between the transformed Radar points and their corresponding LiDAR points is computed as the AE.
>
> The V2X‑Radar dataset shows LiDAR–Camera RPE of 0.27 - 0.33 px (<0.5 px acceptable) and LiDAR–4D Radar AE of 2.23 - 2.35 cm (<5 cm acceptable).
>
> | Platform     | Sensor             | RPE (pixel)↓ | AE (cm)↓ |
> | ------------ | ------------------ | ----------- | ------- |
> | Vehicle-side | LiDAR - CAM        | 0.33        | /       |
> |              | LiDAR - 4D Radar   | /           | 2.35    |
> | Roadside     | LiDAR - CAM\_LEFT  | **0.26**        | /       |
> |              | LiDAR - CAM\_FRONT | 0.27        | /       |
> |              | LiDAR - CAM\_RIGHT | 0.30       | /       |
> |              | LiDAR - 4D Radar   | /           | **2.23**    |
>
> ### **[W2]:** The paper claims 4D Radar improves adverse‑weather perception but offers no ablation evidence.
>
> Thanks for your suggestions. We added ablation studies on vehicle-side frames collected in adverse weather (rain, fog, snow), comparing **LiDAR-only**, **4D Radar-only**, and **LiDAR–4D Radar fusion** models, as shown below. Our findings are as follows:
>
> 1. 4D Radar, while largely behind LiDAR in normal conditions, outperforms LiDAR by ~1–2% mAP under adverse weather, highlighting its robustness to environmental degradation.
>
> 2. The fusion model consistently delivers the best performance across all scenarios, demonstrating the complementary strengths of LiDAR and 4D Radar.
>
> These results highlight 4D Radar’s key role in enhancing perception robustness under adverse weather.
>
> | Method      | Modality         | Veh.(IoU = 0.5)↑ |          |      | Ped.(IoU = 0.25)↑ |          |      | Cyc.(IoU = 0.25)↑ |          |      |
> | ----------- | ---------------- | ------------------- | -------- | ---- | ----------------------- | -------- | ---- | -------------------- | -------- | ---- |
> |             |                  | Easy                | Mod. | Hard | Easy                    | Mod. | Hard | Easy                 | Mod. | Hard |
> | Pointpillar | LiDAR            | 47.23                  | 43.12                      | 42.15                  | 21.12                      | 17.52                         | 16.87                     | 29.32                  | 25.34                       | 24.81                  |
> | Pointpillar | 4D Radar         | 48.35                  | 44.80                      | 43.91                  | 20.14                      | 16.54                         | 16.20                     | 27.85                  | 23.66                       | 22.68                  |
> | M2-Fusion   | LiDAR + 4D Radar | **53.61**                  | **50.18**                      | **49.72**                  | **25.11**                      | **20.53**                         | **19.59**                     | **33.42**                  | **28.24**                       | **27.12**                  |
>
> ### **[W3]:** Section 5.1 says "samples are selected at random," but it’s unclear if by sequence or frame; frame‑level sampling may undermine generalization.
>
> Thanks for your helpful comments. Sampling is done at the **sequence level, not individual frames**: each long recording is split into 30‑frame segments, which are kept intact when assigned to training or validation. This avoids "frame‑level leakage" and maintains a clear domain gap for evaluating generalization.
>
> ### **[W4 & Q4, Q5]:** The labeling and anonymization processes are not described in detail.
>
> Thanks for your insightful comments. And we add the relevant details as follows:
>
> 1. **Labeling Process.** All data were annotated by a **professional labeling company** using an **"auto‑labeling + manual refinement"** workflow: automated tools generated initial labels, which were reviewed and corrected by human annotators. Then, we enforced **strict quality control**, with multiple revision rounds to ensure accuracy and reliability of the final labels.
> 2. **Anonymization Process.** All privacy‑sensitive information (e.g., faces, license plates) was anonymized via **"model‑based detection + manual verification"**: a deep‑learning model blurred sensitive areas, followed by frame‑by‑frame review to guarantee complete and compliant anonymization.
>
> ### **[W5 & Q2, Q3]:** Fig. 5 is hard to interpret, such as scale inconsistency and unclear counts.
>
> Thanks for your constructive comments. Our clarifications are as follows.
>
> 1. **Scale Inconsistency.** We agree this could mislead the day–night sample ratio and will unify subplot scales in the final paper for consistent comparison.
>
> 2. **Unclear Counts.** The correct case is the second option (~10^5 daytime and ~10^4 nighttime detections). We’ll adjust the figure to display daytime and nighttime bars side‑by‑side for clarity.
>
> ### **[W6]:** Tables could benefit from highlighting the data.
>
> Thanks for your valuable suggestions. In the final version, we will highlight key data points through boldface to improve readability and allow important comparisons to stand out more clearly.
>
> ### **[W7 & S3]:**  The conclusion repeats the abstract/intro instead of stressing key insights.
>
> Thanks for your insightful suggestions. We have revised the conclusion as follows and We will update the conclusion in the final version.
>
> In this work, we introduced V2X-Radar, the first large-scale real-world multi-modal dataset featuring 4D Radar for cooperative perception. Our dataset focuses on challenging intersection scenarios and provides data collected under diverse times and weather conditions. Beyond releasing a dataset and benchmarks, our study revealed **two key insights** for the research community:  (1) **Severe performance degradation under asynchronous communication settings**. This finding exposes a critical weakness in the delay robustness of current cooperative perception methods.  (2) **The distinctive advantage of 4D Radar**, which delivers reliable perception in adverse weather and serves as a valuable complement to LiDAR- and camera-based approaches. By releasing **V2X-Radar**, we not only fill the **4D Radar gap** in cooperative perception research but also offer a platform for studying these challenges and validating solutions. We hope this dataset and benchmark will spark future work on **delay-tolerant cooperative perception models**, **robust cross-modal fusion strategies**.
>
> ### **[W8]:** Some minor errors exist.
>
> Thanks for highlighting these minor issues. All corrections will be reflected in the revised manuscript.
>
> ### **[Q1]:** Annotations cover only 5 classes — will you expand or add finer‑grained labels like segmentation?
>
> Thanks for your valuable comments. We agree the five current categories could be expanded. We are **currently** developing an **Occupancy Prediction** task on the V2X‑Radar dataset, framing it as **voxel‑level segmentation** with added static classes (e.g., drivable area, cones, buildings, trees, curbs), enabling richer perception of both dynamic and static elements.
>
> ### **[S1]:** The LiDAR detection baselines are somewhat dated; adding newer methods would strengthen the experiments.
>
> Thanks for you insightful suggestion. We acknowledge some LiDAR detection baselines were dated and have added evaluations of recent state‑of‑the‑art methods (e.g., SQDNet, ACM MM’24; Fade3D, IEEE T‑ITS’25) to ensure a more complete and up‑to‑date benchmark.
> > V2X-Radar-I:
> | Method           | Veh.(IoU = 0.5)↑ |          |      | Ped.(IoU = 0.25)↑ |          |      | Cyc.(IoU = 0.25)↑ |          |      |
> | --------------- | ------------------- | -------- | ---- | ----------------------- | -------- | ---- | -------------------- | -------- | ---- |
> |                           | Easy                | Mod. | Hard | Easy                    | Mod. | Hard | Easy                 | Mod. | Hard |
> | SQDNet  | **95.10** | **86.94** | **86.94** | **88.47** | **84.19** | **84.19** | **95.43** | **87.59** | **87.59** |
> | Fade3D  | 90.43 | 81.72 | 81.72 | 82.85 | 79.07 | 79.07 | 90.76 | 82.17 | 82.17 |
>
> > V2X-Radar-V:
> | Method           | Veh.(IoU = 0.5)↑ |          |      | Ped.(IoU = 0.25)↑ |          |      | Cyc.(IoU = 0.25)↑ |          |      |
> | --------------- | ------------------- | -------- | ---- | ----------------------- | -------- | ---- | -------------------- | -------- | ---- |
> |                           | Easy                | Mod. | Hard | Easy                    | Mod. | Hard | Easy                 | Mod. | Hard |
> | SQDNet  | **90.65** | **85.10** | **85.10** | **72.85** | **65.55** | **65.55** | **91.85** | **82.95** | **82.95** |
> | Fade3D  | 85.77 | 78.95 | 78.95 | 65.92 | 60.94 | 60.94 | 88.29 | 75.80 | 75.80 |
>
> ### **[S2]:** A visualization of the coordinate systems and their relationships would aid reader understanding.
>
> Thanks for you valuable suggestion. We agree a visualization of the coordinate systems would improve clarity. Due to rebuttal media limits, we can’t include it here, but will add clear figures in the final paper.

---

> > ### Comment · Reviewer_o696 · 2025-08-05
> >
> > I appreciate the authors’ efforts in the rebuttal. My concerns have been resolved, and I am glad to see the new experiments. I hope these results will be properly included in the revised version. Therefore, I will raise my score accordingly.

---

> > > ### Author Response · Authors · 2025-08-06
> > > **Thank You for Reading and Consideration**
> > >
> > > Dear Reviewer o696,
> > >
> > > Thanks for your positive response and raised score. We’re glad the clarifications addressed your concerns and will reflect them in the final version.
> > >
> > > Best regards,
> > >
> > > Authors

---

### Official Review · Reviewer_8BTx · 2025-07-03

**Rating:** 5
**Confidence:** 4

**Summary:**

This paper presents V2X-Radar, the first large-scale real-world multi-modal dataset featuring 4D radar for cooperative perception. Unlike existing cooperative perception datasets that focus solely on cameras and LiDAR, this work incorporates 4D radar to enhance robustness in adverse weather conditions. The dataset comprises 20K LiDAR frames, 40K camera images, and 20K 4D radar data with 350K annotated 3D bounding boxes across five categories, collected under diverse weather and lighting conditions. It is organized into three sub-datasets (V2X-Radar-C, V2X-Radar-I, V2X-Radar-V) supporting cooperative perception, roadside perception, and single-vehicle perception research.

**Dataset Code Accessibility:**

Yes

**Ethical Considerations:**

No, there are no or only very minor ethics concerns

**Final Justification:**

After reviewing the authors' rebuttal and additional experiments, I would suggest Accept.

The authors successfully addressed my main concerns. The Doppler ablation demonstrates clear value, and the 100-400ms delay analysis provides important practical insights. The dataset scale is comparable to other V2X datasets, with 540K unlabeled frames to be released. While geographic limitations remain, this is the first large-scale V2X dataset with 4D radar, filling a critical gap in weather-robust perception research. The novel contribution and thorough experimental validation outweigh the remaining limitations.

**Limitations Weaknesses:**

1. Dataset Scale and Scope: The 20K-40K frames are relatively small compared to existing large datasets like nuScenes or Waymo. Data collection limited to specific Chinese intersections raises generalizability concerns for different regions and traffic patterns.

2. Experimental Design: The use of fixed 100ms delays (Section 4.2) does not capture real C-V2X communication dynamics. The focus solely on 3D object detection excludes other critical autonomous driving tasks like tracking and prediction.

3. Analysis Depth: The paper lacks specific analysis of how 4D radar's Doppler information contributes to performance improvements. The absence of ablation studies makes it difficult to quantify the actual benefits of 4D radar inclusion.

**Strengths Contributions:**

1. 4D Radar Dataset Contribution: This work provides the first large-scale real-world cooperative perception dataset incorporating 4D radar. As shown in Table 1, existing datasets focus solely on cameras and LiDAR, while 4D radar has been limited to single-vehicle datasets. This fills a research gap by adding Doppler velocity information for dynamic objects.

2. Technical Implementation: The authors achieve precise temporal synchronization within 20ms using PTP hardware triggers. Figure 3 demonstrates accurate sensor calibration, and clear coordinate system definitions enhance dataset usability.

3. Real-World Data Collection: The dataset comprises actual driving scenarios across diverse weather and lighting conditions, focusing on challenging intersection scenarios that expose single-vehicle perception limitations.

4. Comprehensive Benchmarking: The paper evaluates state-of-the-art algorithms for single-agent and cooperative 3D object detection. Tables 3-5 compare LiDAR, camera, and 4D radar methods, while Figure 6 quantifies communication delay impacts on performance.

---

> ### Author Rebuttal · Authors · 2025-07-30
>
> Thank you for your constructive review comments and insightful questions! We have provided responses below to address each of your concerns.
>
> ### **[W1]**: Limited Dataset Scale and Scope.
>
> Thanks for the question! The concern regarding the dataset’s scale and regional scope is stated as follows:
>
> 1. **Dataset Scale:** High-quality annotation is costly for academic institutions, so we currently release 20K–40K labeled frames. While smaller than vehicle-centric datasets like Waymo and nuScenes, this scale is **comparable to leading V2X datasets** (e.g., DAIR-V2X [A]: 40K, V2X-Seq [B]: 40K, V2V4Real [C]: 20K–40K, TUMTraf-V2X [D]:  2K–5K, RCooper [E]: 30K–50K). Importantly, we have collected **15+ hours (~540K frames)** of raw data, including **long-tail scenarios** such as truck occlusions and pedestrians suddenly appearing ("ghosting" events). Once the necessary privacy filtering is complete, we will **release these unlabeled data** to support semi-/unsupervised research, further improving robustness and generalization.
>
> 2. **Dataset Scope:** we acknowledge that the collected data is currently concentrated in certain intersections in China. Although the geographic coverage is limited, the dataset still spans **diverse traffic densities, varied weather conditions (rain, fog, snow), and both day and night settings**, providing a strong foundation for research on cross-scenario robustness. In future, we plan to **pursue cross‑institutional collaborations** to further enhance the dataset’s scope and value.
>
> ### **[W2]**: Experimental design assumes a fixed 100 ms delay and limits scope to 3D detection.
>
> Thank you for raising these important points. The concerns are addressed as follows:
>
> 1. **Fixed 100 ms Delay:** We acknowledge that using only a fixed 100 ms delay has limitations in representing real‑world C‑V2X communication dynamics. To address this, we followed the asynchronous delay design adopted in DAIR‑V2X [A] and V2V4Real [C] and conducted additional experiments under *100ms*, **200 ms**, **300 ms**, and **400 ms** delays. Beyond the **qualitative performance curves** already shown in Fig. 6 of the manuscript, we now include **quantitative results** (see the table below), which will be integrated into the final version to better capture the impact of variable delays on model performance.
> | Method      | M | Sync. (0ms) ↑ | Async. (100ms)↑ | Async. (200ms)↑  | Async. (300ms)↑ | Async. (400ms)↑ | AM (MB)↓ |
> |-------------|---|-------------|----------------|----------------|----------------|----------------|--------|
> | No Fusion   | L | 21.44       | 21.44          | 21.44          | 21.44          | 21.44          | **0.00**   |
> | Late Fusion | L | 54.38       | 47.84          | 42.91          | 40.62          | 38.66          | 0.00   |
> | F-Cooper    | L | 60.27       | 55.56          | 48.10          | 45.68          | 44.30          | 1.25   |
> | CoAlign     | L | 66.42       | 58.43          | 54.60          | 51.54          | 49.82          | 0.25   |
> | HEAL        | L | **67.75**       | **60.75**         | **56.57**          | **53.12**          | **52.17**          | 0.25   |
> | No Fusion   | C | 3.45        | 3.45           | 3.45           | 3.45           | 3.45           | 0.00   |
> | Late Fusion | C | 12.82       | 11.08          | 10.00          | 9.58           | 9.12           | 0.00   |
> | F-Cooper    | C | 17.51       | 15.61          | 14.01          | 13.27          | 12.87          | 1.25   |
> | CoAlign     | C | 19.33       | 17.33          | 16.04          | 15.00          | 14.50          | 0.25   |
> | HEAL        | C | 19.58       | 17.42          | 16.45          | 15.35          | 15.08          | 0.25   |
> | No Fusion   | R | 6.98        | 6.98           | 6.98           | 6.98           | 6.98           | 0.00   |
> | Late Fusion | R | 10.95       | 9.76           | 8.98           | 8.18           | 7.79           | 0.00   |
> | F-Cooper    | R | 14.23       | 12.68          | 11.81          | 10.79          | 10.12          | 1.25   |
> | CoAlign     | R | 15.59       | 13.71          | 13.10          | 12.10          | 11.08          | 0.25   |
> | HEAL        | R | 15.82       | 13.97          | 13.45          | 12.40          | 11.25          | 0.25   |
>
> 2. **Scope beyond 3D Object Detection:** The V2X‑Radar dataset already **contains Tracking ID annotations**, providing a strong foundation for tasks such as tracking and prediction. In this rebuttal, we present an **initial tracking benchmark** (see table below) and will develop a more comprehensive, task‑agnostic baseline in future work to better support tracking, prediction, and broader V2X applications.
> | Method      | M  | AMOTA↑ | AMOTP↑ | sAMOTA↑ | MOTA↑ | MT↑   | ML↓   |
> |-------------|----|--------|--------|--------|-------|-------|-------|
> | No Fusion   | L  | 13.79  | 24.29  | 35.79  | 26.37 | 20.79 | 65.25 |
> | Late Fusion | L  | **29.78**  | **50.88**  | **64.78**  | **52.66** | **44.78** | **33.15** |
> | No Fusion   | C  | 3.60   | 9.40   | 21.70  | 10.63 | 8.33  | 89.33 |
> | Late Fusion | C  | 10.43  | 19.50  | 32.43  | 21.53 | 17.50 | 72.21 |
> | No Fusion   | R  | 4.22   | 12.50  | 22.22  | 13.85 | 9.22  | 85.02 |
> | Late Fusion | R  | 9.84   | 18.19  | 31.84  | 20.43 | 16.84 | 75.87 |
>
> ### **[W3]**:  The paper lacks ablations clarifying Doppler’s role in 4D radar performance gains.
>
> Thanks for the insightful comment! Prior research has underscored its significance — for example, **VOD [F]** demonstrated that incorporating the Doppler dimension (+1D) enhances the separation of moving objects from static background clutter, thereby substantially improving the detection of moving objects. To substantiate this point, we have **conducted ablation experiments** on samples collected under **adverse weather conditions** (rain, heavy fog, and snow), as summarized in the table below. The results reveal several key findings:
>
> 1. 4D Radar with Doppler consistently surpasses its Doppler‑free counterpart across all object categories, with particularly pronounced improvements for vehicle detection (+3.41 AP).
>
> 2. Relative to LiDAR, 4D Radar with Doppler narrows the performance gap and even achieves superior results in adverse weather conditions, such as Vehicle (43.12 vs. 44.80).
>
> Collectively, these findings provide compelling empirical evidence that **Doppler information is not merely an auxiliary input but a critical sensing dimension**, reinforcing the role of 4D Radar as a robust modality for perception under adverse weather conditions.
> | Modality           | Veh. (IOU = 0.5)↑|        |        | Ped. (IOU = 0.25)↑|        |        | Cyc. (IOU = 0.25)↑|        |        |
> |--------------------|---------------------|--------|--------|-------------------------|--------|--------|----------------------|--------|--------|
> |                    | Easy                | Mod. | Hard   | Easy                   | Mod. | Hard   | Easy                 | Mod. | Hard   |
> | LiDAR                 | 47.23                  | 43.12                      | 42.15                  | **21.12**                      | **17.52**                         | **16.87**                     | **29.32**                  | **25.34**                       | **24.81**                  |
> | 4D Radar w/o Doppler  | 45.22                  | 41.39                      | 40.92                  | 19.98                      | 14.30                         | 14.24                     | 24.68                  | 21.52                       | 20.56                  |
> | 4D Radar w/ Doppler   | **48.35**                  | **44.80**                      | **43.91**                  | 20.14                      | 16.54                         | 16.20                     | 27.85                  | 23.66                       | 22.68                  |
>
> ### **Reference:**
>
> [A] Yu H, Luo Y, Shu M, et al. DAIR-V2X: A large-scale dataset for vehicle-infrastructure cooperative 3d object detection[C]//Proceedings of the IEEE/CVF conference on computer vision and pattern recognition. 2022: 21361-21370.
>
> [B] Yu H, Yang W, Ruan H, et al. V2x-seq: A large-scale sequential dataset for vehicle-infrastructure cooperative perception and forecasting[C]//Proceedings of the IEEE/CVF Conference on Computer Vision and Pattern Recognition. 2023: 5486-5495.
>
> [C] Xu R, Xia X, Li J, et al. V2v4real: A real-world large-scale dataset for vehicle-to-vehicle cooperative perception[C]//Proceedings of the IEEE/CVF conference on computer vision and pattern recognition. 2023: 13712-13722.
>
> [D] Zimmer W, Wardana G A, Sritharan S, et al. Tumtraf v2x cooperative perception dataset[C]//Proceedings of the IEEE/CVF conference on computer vision and pattern recognition. 2024: 22668-22677.
>
> [E] Hao R, Fan S, Dai Y, et al. Rcooper: A real-world large-scale dataset for roadside cooperative perception[C]//Proceedings of the IEEE/CVF conference on computer vision and pattern recognition. 2024: 22347-22357.
>
> [F] Palffy A, Pool E, Baratam S, et al. Multi-class road user detection with 3+ 1d radar in the view-of-delft dataset[J]. IEEE Robotics and Automation Letters, 2022, 7(2): 4961-4968.

---

> > ### Comment · Reviewer_8BTx · 2025-08-07
> >
> > The rebuttal adequately addresses my main concerns. I am inclined to maintain or slightly increase my rating based on these clarifications, contingent on incorporating the promised additions in the final version.

---

> > > ### Author Response · Authors · 2025-08-07
> > > **Thank You for Reading and Consideration**
> > >
> > > Dear Reviewer 8BTx,
> > >
> > > Thank you for your positive response. We’re glad our clarifications addressed your concerns and will ensure they are incorporated into the final version.
> > >
> > > Best regards,
> > >
> > > Authors

---

### Note · Authors · 2025-08-12

Dear AC and Reviewers,

We sincerely thank all reviewers for recognizing the novelty of V2X-Radar. Our rebuttal addressed key concerns—dataset coverage, calibration precision, and robustness evaluation—leading to raised scores from multiple reviewers and a clear trend toward acceptance.

**Reviewer-recognized strengths:**

- **Novelty:** First large-scale cooperative perception dataset with 4D Radar for robust perception in adverse conditions (8BTx, o696, CafA).
- **Rich Real-world Data:** Diverse weather, lighting, and complex intersections (8BTx, wjZP).
- **Flexible Dataset Structure:** Single-agent & cooperative tasks across multiple modalities (o696, wjZP).
- **Technical Rigor:** Precise synchronization and thorough sensor calibration ensuring usability and reproducibility (8BTx, o696).
- **Comprehensive Benchmarking:** Comprehensive evaluations establishing clear baselines and revealing current method limitations (8BTx, o696, wjZP).
- **Clarity & Accessibility:** Clear writing, informative figures/tables, open-source release (o696, CafA).

**Key revisions & clarifications:**

- **Dataset Scale & Scope:**  Comparable to DAIR-V2X/V2V4Real, with additional raw data planned for semi-/unsupervised research.
- **Geographic Coverage & Diversity:** Collected in one region but covering 10+ location types with diverse traffic and weather, with planned expansion via cross-institutional collaborations.
- **New Experiments:**
  - Delay robustness (100–400 ms) → confirmed need for delay-tolerant designs.
  - Initial tracking benchmarks using existing ID annotations.
  - Doppler ablation (+3.41 AP in adverse weather).
  - adverse weather analysis (4D Radar > LiDAR; fusion best).
  - updated baselines (SQDNet, Fade3D).
- **Calibration Metrics:** Reported LiDAR–Camera RPE (<0.33 px) and LiDAR–Radar AE (<2.35 cm), confirming high-precision alignment.
- **Labeling & Anonymization:** Clarified auto-label + manual refinement pipeline with strict QC; model-assisted anonymization verified by human reviewers.

With clear novelty, technical rigor, and comprehensive benchmarks—and with all major concerns addressed—we believe V2X-Radar is positioned to be a long-term, high-impact community resource for cooperative perception research.


**Best regards,**

Authors

---

### Decision · Program_Chairs · 2025-09-18

**Decision:**

Accept (spotlight)

**Comment:**

This paper introduces V2X-Radar, the first large-scale multimodal dataset incorporating 4D radar for cooperative perception. The dataset is well-structured, includes diverse conditions, and provides comprehensive benchmarks with strong baselines. Reviewers consistently praised its novelty, careful sensor synchronization, and potential impact on robust perception research.

While some concerns were raised regarding dataset scale, geographic coverage, and the need for further weather-based evaluation or additional tasks, the authors effectively addressed these points during the rebuttal (e.g., Doppler ablation, delay analysis, release of unlabeled frames). Given its clear novelty, thorough experimental validation, and strong community value, all reviewers converged on recommending acceptance.